# Decoding the genomic landscape of chromatin-associated biomolecular condensates

Zhaowei Yu[1,5], Qi Wang[1,5], Qichen Zhang[2,3,5], Yawen Tian[3], Guo Yan[3], Jidong Zhu[4], Guangya Zhu [3] ✉ & Yong Zhang [1] ✉

Biomolecular condensates play a significant role in chromatin activities, primarily by concentrating and compartmentalizing proteins and/or nucleic acids. However, their genomic landscapes and compositions remain largely unexplored due to a lack of dedicated computational tools for systematic identification in vivo. To address this, we develop CondSigDetector, a computational framework designed to detect condensate-like chromatin-associated protein co-occupancy signatures (CondSigs), to predict genomic loci and component proteins of distinct chromatin-associated biomolecular condensates. Applying this framework to mouse embryonic stem cells (mESC) and human K562 cells enable us to depict the high-resolution genomic landscape of chromatin-associated biomolecular condensates, and uncover both known and potentially unknown biomolecular condensates. Multi-omics analysis and experimental validation further verify the condensation properties of CondSigs. Additionally, our investigation sheds light on the impact of chromatin-associated biomolecular condensates on chromatin activities. Collectively, CondSigDetector provides an approach to decode the genomic landscape of chromatin-associated condensates, facilitating a deeper understanding of their biological functions and underlying mechanisms in cells.

Over the last decade, there has been growing appreciation for the biological role of biomolecular condensates, which are membrane-less compartments that compartmentalize and concentrate specific proteins and/or nucleic acids[1,2]. Liquid-liquid phase separation (LLPS) has been proposed as a key organizing principle of biomolecular condensates, driven by weak, multivalent, and highly collaborative molecular interactions[2]. The molecular interactions inside biomolecular condensates usually involve diverse collaborative components that can be categorized into two main groups: scaffolds and clients. Scaffolds drive the formation of condensates, while clients participate by binding to scaffolds[3–6]. Biomolecular condensates are implicated in

various cellular functions, and their aberrations are associated with numerous diseases[1,7]. Recently, growing evidences have demonstrated the widespread existence and functional significance of chromatin-associated biomolecular condensates. Many chromatin-associated processes, such as DNA replication[8], DNA repair[9], transcription control[10–13], and chromatin organization[14–17], have been found to take place within biomolecular condensates at chromatin[18] (Supplementary Data 1).

Understanding chromatin-associated biomolecular condensates, including their genomic loci and collaborative components, is crucial for elucidating their impact on chromatin activities. Although some

[1]State Key Laboratory of Cardiovascular Diseases and Medical Innovation Center, Institute for Regenerative Medicine, Department of Neurosurgery, Shanghai East Hospital, Shanghai Key Laboratory of Signaling and Disease Research, Frontier Science Center for Stem Cell Research, School of Life Sciences and Technology, Tongji University, Shanghai 200092, China. [2]Pancreatic Intensive Care Unit, Changhai hospital, Naval Medical University, Shanghai 200433, China. [3]Lingang Laboratory, Shanghai 200031, China. [4]Etern Biopharma, Shanghai 201203, China. [5]These authors contributed equally: Zhaowei Yu, Qi Wang, Qichen Zhang. ✉e-mail: zhugy@lglab.ac.cn; yzhang@tongji.edu.cn

chromatin-associated biomolecular condensates have been linked to well-characterized chromatin states, such as super-enhancer[10,11] and heterochromatin[15–17], these connections have generally been reported without comprehensive associations with genome-wide loci, except for a few loci of interest validated by low-throughput experiments. Until now, the genomic landscape of chromatin-associated biomolecular condensates has remained poorly understood. However, no genomic approach has been designed yet to capture the comprehensive genomic landscape of chromatin-associated biomolecular condensates, primarily due to the following challenges. First, the complexity of biomolecular condensates arising from their diverse components[18] and context-specific molecular collaborations among these components along the chromatin[13], making it difficult to systematically capture chromatin-associated biomolecular condensates by targeting a single factor. Second, even for chromatin-associated protein (CAP) with experimental evidence of condensation[3–6,19], distinguishing its condensation-associated binding sites from non-associated binding sites in individual datasets is not a straightforward task.

With the rapid accumulation of CAP occupancy profiles and proteome-scale characterization of condensation potential, it is now possible to overcome the above challenges of decoding the genomic landscape of chromatin-associated biomolecular condensates by integrating multi-dimensional data. In this study, we introduce CondSigDetector, a computational framework that systematically predicts chromatin-associated biomolecular condensates. This framework overcomes the two challenges mentioned above by utilizing topic modeling to detect genome-wide context-dependent collaborations among CAPs possessing high condensation potential from hundreds of CAP occupancy profiles. These collaborations along the chromatin are termed Condensate-like chromatin-associated protein co-occupancy Signatures (CondSigs). The framework not only identifies the collaborative components of distinct biomolecular condensates, but also assigns them to the associated genomic loci. We apply this computational framework to two cell types with abundant ChIP-seq data, and predict hundreds of chromatin-associated biomolecular condensates, along with their genomic loci, which are supported by multi-omics data and experimental evidences. CondSigDetector is a computational framework for decoding the genomic landscape of chromatin-associated biomolecular condensates, providing a valuable resource for investigating the functional effects and underlying mechanisms of chromatin-associated biomolecular condensates on chromatin activities.

## Results

### Overall design of CondSigDetector

By integrating ChIP-seq datasets of hundreds of CAPs in the same cell type, we observed frequent co-occupancy of CAPs across the genome (Supplementary Fig. 1a, b). However, co-occupancy events could not be fully explained by DNA binding motifs or chromatin accessibility (Supplementary Fig. 1c–f), two known determinants of CAP co-occupancy events[20]. Furthermore, neither the presence of histone modifications nor physical protein-protein interactions could fully account for all co-occupancy events (Supplementary Fig. 1e–h). This suggests that alternative mechanisms may also be responsible for organizing genome-wide co-occupancy events of CAPs. Biomolecular condensation at chromatin may explain a part of such events, as biomolecular condensates are thought to be mediated by collaborations of components[2], and condensations of CAPs have been reported to influence their chromatin occupancy[10,21]. This suggests that specific CAP co-occupancy events could act as signatures of chromatin-associated biomolecular condensates. Consequently, identifying specific CAP co-occupancy patterns, particularly those mediated by CAPs with high condensation potentials, offers a powerful approach to predict the presence of chromatin-associated biomolecular condensate in the genome.

In this study, we aim to predict chromatin-associated biomolecular condensates by detecting genome-wide context-dependent collaborations of CAPs with high condensation potential, termed CondSig. We developed a computational framework, CondSigDetector, to systematically detect CondSigs by integrating hundreds of ChIP-seq datasets and condensation-related characterizations of CAPs (Fig. 1). CondSigDetector comprises three steps: data processing, co-occupancy signatures identification, and condensation potential filtration.

In the first step, the input data, *i.e.*, the collected ChIP-seq profiles of all CAPs from an identical cell type, is converted into an occupancy matrix at genome-wide consecutive bins. To address the sparsity of this matrix, CondSigDetector applies an iterative segmentation method for each target CAP, which segments the entire occupancy matrix into smaller sub-matrices (see Methods for details). This segmentation approach can enhance the detection of CAP collaborations in local contexts by substantially increasing the occurrence frequency of co-occupancy events within the sub-matrices (Supplementary Fig. 2a, b).

In the second step, CondSigDetector utilizes a topic model to identify co-occupancy signatures of CAPs, representing frequent CAP collaborations, from the sub-matrices. Given the significant differences in co-occupancy frequencies between promoter and non-promoter regions (Supplementary Fig. 1a, b), the sub-matrices are categorized into either promoter or non-promoter groups to identify co-occupancy signatures separately. Within the topic model, each sub-matrix is treated as a set of documents, where each genomic bin represents a document and CAPs occupying the bin are considered as words in the document. Intuitively, the topics learned from topic modeling, which indicate specific word combinations, can be interpreted as co-occupancy signatures of CAPs. Since the number of co-occupied CAPs within a bin is typically sparse (Supplementary Fig. 1a, b), CondSigDetector utilizes the biterm topic model, which outperforms traditional models such as Latent Dirichlet Allocation for short text[22]. It has been confirmed that the co-occupancy signatures of CAPs derived from the biterm topic model exhibit high topic coherence and repeatability among replicates (see Methods for details; Supplementary Fig. 2c–f).

In the third step, CondSigDetector predicts CondSigs by evaluating the condensation potential for each co-occupancy signature of CAPs. For each genomic bin, 6 condensation-related features are calculated: the fraction of occupied CAPs with reported LLPS capacity, the fraction of occupied CAPs co-occurring in the same membrane-less organelle (MLO), the fraction of occupied CAPs with predicted intrinsically disordered regions (IDRs), the fraction of occupied CAP pairs having protein-protein interactions (PPIs), the fraction of occupied CAPs predicted as RNA-binding proteins (RBPs), and the RNA-binding strength (RBS) of the bin. Intuitively, for a co-occupancy signature of CAPs, higher values of these condensation-related features at signature-positive bins indicate a greater condensation potential. Co-occupancy signatures with at least three condensation-related features strongly and positively correlated with their presence are identified as CondSigs (see Methods for details). It has been confirmed that the CondSigDetector can successfully recover most of the identified CondSigs when specific transcription factor families were removed from the full dataset (Supplementary Fig. 2g, h), which demonstrates the robustness of the computational framework. Finally, CondSigDetector eliminates redundant CondSigs containing similar components.

### Identification of CondSigs in mouse and human cell lines

CondSigDetector was applied to two cell types with abundant ChIP-seq data: mESC and human K562 cell line, to identify CondSigs. After

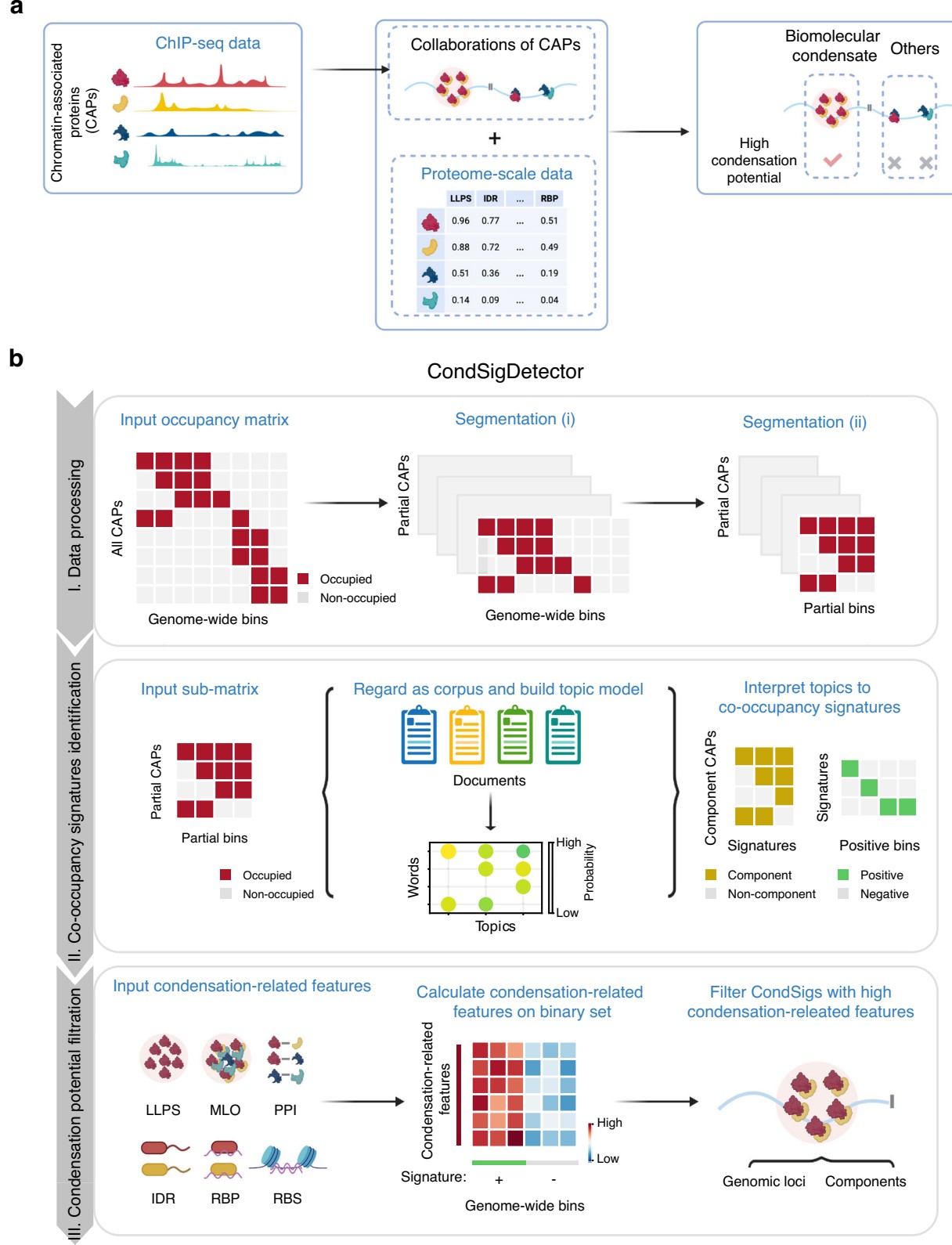

**Fig. 1 | Overall design of CondSigDetector. a** A schematic illustrates the prediction of chromatin-associated biomolecular condensates by detecting genome-wide context-dependent collaborations of CAPs with high condensation potential. **b** Workflow of CondSigDetector (see Methods for details). Figure 1 created with BioRender.com released under a Creative Commons Attribution-NonCommercial-NoDerivs 4.0 International license.

stringent quality control, we gathered qualified ChIP-seq data for 189 CAPs in mESC and 216 CAPs in K562 (Supplementary Data 2). Due to the lack of a qualified RNA-binding profile for mESC, the RNA binding strength, one of the condensation-related features, was not included in

mESC. We identified 25 promoter CondSigs and 36 non-promoter CondSigs in mESC (Fig. 2a), along with 75 promoter CondSigs and 93 non-promoter CondSigs in K562 (Supplementary Fig. 3). Additionally, we identified 14,345 promoter CondSig-positive sites and 24,500 non-

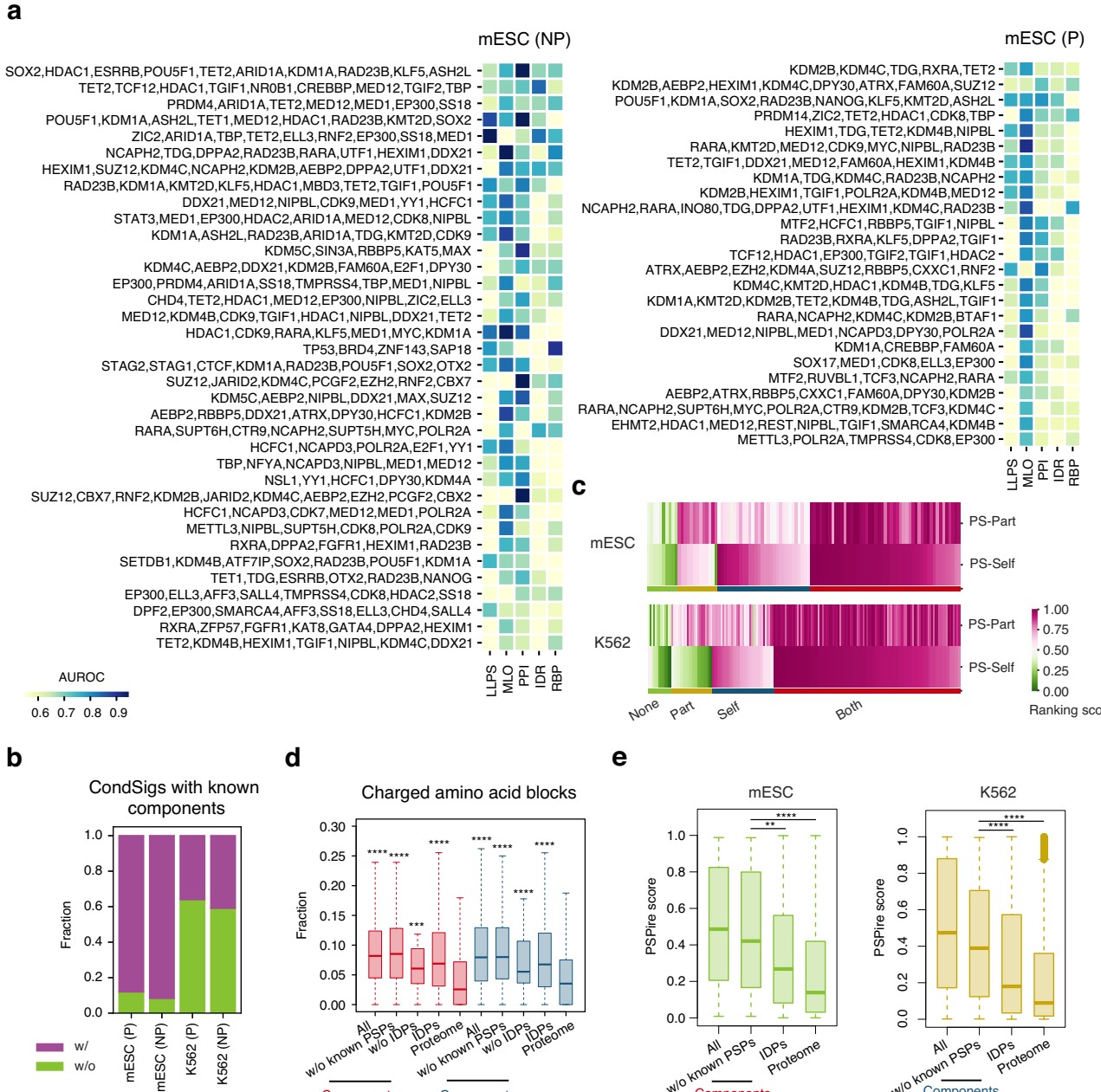

**Fig. 2 | CondSigs in mESC and K562. a** Heatmaps showing identified CondSigs in mESC at non-promoter (NP) and promoter (P) regions. Each row represents a CondSig and the row name indicates the component CAPs of the given CondSig. Each column represents a condensation-related feature and the colours represent AUROC. **b** The stacked bar plots showing the fraction of CondSigs with components in known chromatin-associated biomolecular condensates (Supplementary Data 1). The overlap set of known chromatin-associated biomolecular condensates and known MLOs is not considered in the calculation. **c** Heatmaps showing *k*-means clustering for component CAPs in mESC (left) and K562 (right) with PS-Self and PS-Part ranking score (see Methods for details). Both high PS-Self and PS-Part ranking score refer to scaffolds in phase separation. Four clusters ("Both": both Self and Part, "Self": Self-only, "Part": Part-only, and "None") were shown. **d** Box plots showing component CAPs of CondSigs (all, components without known phase-separation proteins and components without predicted IDR-containing proteins) and all predicted IDR-containing proteins have higher fractions of charged blocks in amino acid sequences than control (mouse or human proteome) in mESC and K562 (see Methods for details about the prediction of IDRs of proteins and the

annotation of charged amino acid blocks). The centre lines in boxes mark the median, the box limits indicate the 25th and 75th percentiles, and the whiskers extend to 1.5 × the interquartile range from the 25th and 75th percentiles. Statistical significances between component CAPs / IDPs and control proteome were evaluated by one-sided Wilcoxon rank sum tests, *** represents *p*-value < 1 × 10⁻³ and **** represents *p*-value < 1 × 10⁻⁴. Sample sizes used to derive statistics are 103, 79, 31, 5835 and 55,260 for mESC and 176, 151, 62, 6062 and 20,594 for K562. **e.** Box plots showing component CAPs (with or without known phase-separating proteins) have higher PSPire scores than all predicted IDPs or the entire proteome. The scores were from PSPire, a machine learning predictor for the precise prediction of phase-separating proteins[27]. The centre lines in boxes mark the median, the box limits indicate the 25th and 75th percentiles, and the whiskers extend to 1.5 × the interquartile range from the 25th and 75th percentiles. Statistical significances were evaluated by two-sided Welch's *t*-tests, ** represents *p*-value < 0.01 and **** represents *p*-value < 1 × 10⁻⁴. Sample sizes used to derive statistics are 101, 78, 5,864 and 21,615 for mESC and 176, 151, 6,096 and 20,296 for K562. Source data are provided as a Source Data file.

promoter CondSig-positive sites in mESC, along with 14,201 and 38,963 CondSig-positive sites in K562. To assess the reliability of identified CondSigs, we examined whether their component CAPs are involved in known chromatin-associated biomolecular condensates. Among the identified mESC CondSigs, 88.0% of promoter and 91.7% of non-promoter CondSigs contain at least one component CAP present in known chromatin-associated biomolecular condensates that do not overlap with input MLOs (Fig. 2b). For example, a non-promoter CondSig contains SS18, SMARCA4 (BRG1), and DPF2, which are three known components of the known SS18 cluster[23] (Fig. 2a, Supplementary Fig. 4a, Supplementary Data 1). In K562 cells, 36.0% of promoter and 40.9% of non-promoter CondSigs have at least one component CAP found in known chromatin-associated biomolecular condensates (Fig. 2b). One example of a non-promoter CondSig includes CBX5 (HP1α), TRIM28 and CBX1 (HP1β) (Supplementary Fig. 4b, Supplementary Data 1), with HP1 and TRIM28 were reported to drive LLPS with H3K9me3-modified chromatin cooperatively[15]. These results provide support for the reliability of the identified CondSigs.

Some component CAPs are found in more than one identified CondSig (Supplementary Fig. 4c, d). For example, DDX21, a DEAD-box RNA helicase known to participate in biomolecular condensate[24], is present in 8 non-promoter CondSigs in mESC. We examined the similarity of present loci between CondSig pairs containing at least one shared component CAP and found that only 0.6% of mESC pairs and 0.9% of K562 pairs had a Jaccard index higher than 0.7. This suggests a high diversity of present loci of identified CondSigs, even when they share some common components. To investigate the potential roles of component CAPs in CondSigs, we classified all predicted component CAPs into four clusters: "both Self and Part", "Self-only", "Part-only", and "none", according to their calculated potentials for self-assembly (PS-Self) or interaction with partners (PS-Part) to undergo phase separation[25] (see Methods for details). 90.2% and 92.6% of component CAPs in mESC and K562 were classified into "both Self and Part", "Self-only" or "Part-only" clusters (Fig. 2c). And the percentages are still high in mESC (90.9%) and K562 (90.8%) when we excluded known MLO memberships from the component CAPs (Supplementary Fig. 4e). Furthermore, we found that component CAPs of CondSigs have a high fraction of charged amino acid blocks (Fig. 2d), which is an important resource for multivalency[26]. After removing phase-separating proteins or predicted IDR-containing proteins, the remaining component CAPs still showed comparable fractions to all IDR-containing proteins and higher fractions than the entire proteome. We also utilized PSPire, a recently developed machine learning predictor designed to integrate residue-level and structure-level features for the precise prediction of phase-separating proteins[27], to examine the phase separation capacities of component CAPs. Remarkably, the component CAPs exhibited higher PSPire scores, which remained elevated after the exclusion of known phase-separating proteins (Fig. 2e). These results suggest that the component CAPs of identified CondSigs have strong capacities to form biomolecular condensates, and may function in a context-dependent manner.

The previous studies demonstrated that biomolecular condensate can form at super-enhancers, i.e., clusters of enhancers densely occupied by the master regulators and mediators, and these condensates can regulate gene transcription by concentrating transcription machinery[10,28]. When comparing the genomic loci of super-enhancers[29] and CondSig-positive sites, we found that 93.8% (743 out of 792) of super-enhancers in mESC and 97.5% (668 out of 685) in K562 overlapped with CondSig-positive sites. Moreover, we evaluated the reliability of CondSig-positive sites by considering sites significantly affected by the inhibition of well-characterized CAPs involved in biomolecular condensates as potential positive markers. We re-analyzed H3K27ac, Pol II ChIP-seq data in mESC following the inhibition of EP300[30], an important chromatin regulator associated with biomolecular condensates in mESC[31]. After treatment with A-485, the EP300

inhibitor, we observed a substantial decrease in H3K27ac and Pol II signals at EP300 peaks. Specifically, 51.2% (3652 out of 7126) of CondSig-positive EP300 peaks exhibited a significant decrease in H3K27ac signals, while only 20.8% (1231 out of 5906) CondSig-negative EP300 peaks exhibited a similar significant decrease (Supplementary Fig. 4f). For Pol II, the percentages were 42.7% and 21.9%. The results indicate that transcription regulation of CondSig-positive sites was greatly affected by EP300, a condensate-involved CAP, supporting the accuracy of identified CondSig-positive sites.

## Chromatin properties of identified CondSigs

To investigate the chromatin features of identified CondSigs, we first analyzed the concentration levels of the component within CondSigs by calculating ChIP-seq signal strength for each component. We divided the ChIP-seq peaks of each component CAP into CondSig-positive peaks or -negative peaks based on their overlap with sites where the given CAP was predicted as a component of any CondSigs (see Methods for details), and compared their ChIP-seq signals. As shown in Fig. 3a, most component CAPs displayed significantly higher signal strength at CondSig-positive peaks in mESC, indicating that CondSigs can concentrate their components at target genomic loci. For example, CTCF, a CAP involved in chromatin insulation[32], exhibited significantly higher signal strength at CondSig-positive CTCF peaks. To investigate the biological functional effect of CTCF concentration, we re-analyzed Micro-C data in mESC[33] and found that CondSig-positive CTCF peaks exhibited significantly higher boundary strength than CondSig-negative CTCF peaks (Fig. 3b), suggesting that CTCF concentration contributes to enhanced chromatin insulation activity. We then merged the adjacent ChIP-seq peaks to obtain domains for each component CAP (see Methods for details), and compared the width distributions of CondSig-positive and -negative domains. As shown in Fig. 3c, CondSig-positive domains are wider on average for 95.2% and 93.5% of all component CAPs of promoter and non-promoter CondSigs, and the CondSig-positive domains of RUVBL1, TCF3, CTR9, MTF2 and SUPT6H exceeded 10 kb on average. Additionally, we assessed the component concentration levels and domain widths of CondSigs in K562 and found largely consistent results (Supplementary Fig. 5a, b). These results indicate the component concentration properties of CondSigs, which is a basic feature of known chromatin-associated biomolecular condensates[18], and suggested a potential association between biomolecular condensation and stronger effects on chromatin activities.

Based on previous studies that reported spatially proximal chromatin could be involved in the same condensates[34,35], we processed to analyze chromatin contact frequencies within and between CondSig-positive and -negative domains for each component CAP. In order to minimize the impact of distinct width distributions between CondSig-positive and -negative domains, we focused on broad domains (width > 5 kb). We used cohesin ChIA-PET data from mESC[36] to measure chromatin interactions between genomic loci, and found that CondSig-positive domains exhibited significantly higher intra-domain interactions than their CondSig-negative counterparts (Fig. 3d). We further calculated the fractions of domains with chromatin interactions within the same group of domains for each component CAP, and found significantly higher frequencies between CondSig-positive domains compared to CondSig-negative domains (Fig. 3e). For each component CAP presented in both promoter and non-promoter CondSigs, we calculated fractions of domains with chromatin interactions between its promoter and non-promoter domains. Our analysis observed that CondSig-positive domains showed significantly higher frequencies between promoter and non-promoter domains relative to CondSig-negative domains (Fig. 3f). We also utilized Pol II ChIA-PET data[37] to evaluate the chromatin contact frequencies of CondSigs in K562, and observed largely consistent results (Supplementary Fig. 5c–e). These results confirmed that the components of

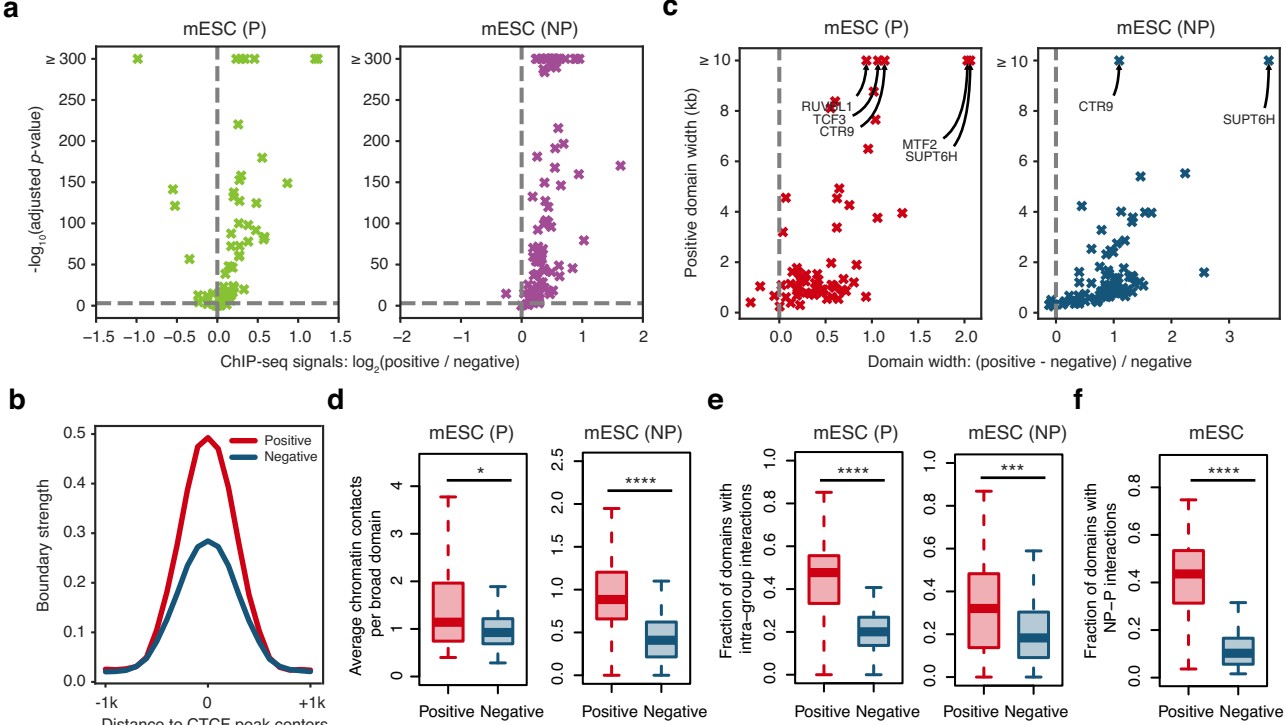

**Fig. 3 | Chromatin properties of predicted chromatin-associated biomolecular condensates. a** Volcano plots showing concentration levels of component CAPs in CondSigs in mESC. *X*-axis represents the log$_2$-transformed fold change of ChIP-seq signals at CondSig-positive peaks compared to CondSig-negative peaks, while *Y*-axis represents the negative log$_{10}$-transformed adjusted *p*-value. Statistical significance between groups was evaluated by a two-sided Welch's *t*-test and the Benjamini-Hochberg (BH) procedure was applied to adjust *p*-values for multiple testing. The vertical dashed line corresponds fold change = 1 and the horizontal dashed line corresponds to *p*-value = 0.001. **b** Line charts showing boundary strength around CondSig-positive and -negative CTCF peaks. Boundary strength was calculated using Micro-C data in mESC from the previous study (GSE130275[33]). **c** Scatter plots showing width comparison of CondSig-positive and -negative domains in mESC. *X*-axis represents the ratio to which the CondSig-positive domain width exceeds the CondSig-negative domain width, and *Y*-axis represents the positive domain width. Component CAPs having CondSig-positive domains exceeding 10 kb on average were labeled. **d** Intra-domain chromatin contacts of CondSig-positive or -negative broad domains in mESC. For each component CAP, an average valid paired-end tags count in each broad domain (> 5 kb) was calculated to represent intra-domain contacts. The centre lines mark the median, the box limits indicate the 25th and 75th percentiles, and the whiskers extend to 1.5 × the interquartile range from the 25th and 75th percentiles. Statistical significance

between groups was evaluated by a one-sided Welch's *t*-test, * represents *p*-value < 0.05 and **** represents *p*-value < 1 × 10$^{-4}$. Sample size used to derive statistics is 61 for promoter regions and 91 for non-promoter regions. Cohesin ChIA-PET data used in the analysis was from the previous study (GSE57913[36]). **e** Box plots showing intra-group chromatin contacts between CondSig-positive or -negative domains in mESC. For each component CAP, the fraction of domains having at least one valid paired-end tag with other intra-group domains was calculated. The centre lines mark the median, the box limits indicate the 25th and 75th percentiles, and the whiskers extend to 1.5 × the interquartile range from the 25th and 75th percentiles. Statistical significance between groups was evaluated by a one-sided Welch's *t*-test, *** represents *p*-value < 1 × 10$^{-3}$ and **** represents *p*-value < 1 × 10$^{-4}$. Sample size used to derive statistics is 61 for promoter regions and 91 for non-promoter regions. **f**. Box plots showing NP (non-promoter)-P (promoter) chromatin contacts between CondSig-positive or -negative domains in mESC. For each component CAP, the fraction of non-promoter domains having at least one valid paired-end tag with its promoter domains was calculated. The centre lines mark the median, the box limits indicate the 25th and 75th percentiles, and the whiskers extend to 1.5 × the interquartile range from the 25th and 75th percentiles. Statistical significance between groups was evaluated by a one-sided Welch's *t*-test, **** represents *p*-value < 1 × 10$^{-4}$. Sample size used to derive statistics is 53. Source data are provided as a Source Data file.

---

identified CondSigs can be concentrated in trans through spatially proximal chromatin. Furthermore, to ensure fair comparison, we refined the criteria for the identification of CondSig-positive and -negative peaks to make they are matched in terms of chromatin accessibility or the number of co-occupied CAPs (see Methods for details). Subsequent comparisons between the refined CondSig-positive and -negative peaks/domains confirmed that CondSig-positive groups consistently exhibit higher signal strength and chromatin contact frequencies (Supplementary Fig. 6, 7).

### Involvement of DDX21 in chromatin-associated biomolecular condensate

Although DDX21 can undergo phase separation and has been reported to participate in nucleolar condensate for Pol I transcription[24,38], additional genomic loci where it may involve into biomolecular condensate remain to be elucidated. In mESC, we identified 10 CondSigs

with DDX21 as a component, with 15,578 DDX21 ChIP-seq peaks as CondSig-positive peaks. To verify the presence of DDX21-associated biomolecular condensate at these genomic loci, we assessed the sensitivity of DDX21 occupancy at these loci to 1,6-hexanediol (1,6-HD), a compound used for disrupting liquid-like biomolecular condensates[39]. Cleavage Under Targets and Release Using Nuclease (CUT&RUN) experiments were conducted for DDX21 in both wild type and 1,6-HD-treated mESC. When all peaks were ranked by the extent to which DDX21 was decreased upon 1,6-HD treatment, CondSig-positive peaks were significantly enriched among those that lost DDX21 (Supplementary Fig. 8a, b), demonstrating the strong effect of biomolecular condensate disruption on CondSig-positive peaks of DDX21. 2,5-HD, while chemically similar to 1,6-HD, is not as strong as 1,6-HD in dissolving the hydrophobicity-dependent condensate[39], so it was considered as a negative control of 1,6-HD. The strong association between the presence of CondSig and the decrease of DDX21 was also

observed when we used 2,5-HD treatment as the control condition of 1,6-HD treatment (Supplementary Fig. 8b–e). These evidences suggested that DDX21 participates in biomolecular condensates at these loci. We further investigated the potential impact of DDX21-associated biomolecular condensates at these genomic loci. We found that target genes of the CondSig-positive peaks of DDX21 displayed significantly higher expression levels than other genes (Supplementary Fig. 8f), suggesting that DDX21-associated biomolecular condensate may enhance the transcription of target genes.

## Confirmation of identified CondSigs

SUPT6H, SUPT5H and CTR9 have been reported to regulate transcription elongation[40,41], but it remains unclear whether these CAPs function in the form of condensate. In mESC, we identified two CondSigs containing all or at least three of SUPT6H, SUPT5H, CTR9, and POLR2A simultaneously (Fig. 2a, Supplementary Fig, 9a). Genomic enrichment analysis found that merged CondSig-positive sites of the two CondSigs were primarily located at promoters and gene bodies (especially at exons), and the associated gene bodies were enriched with H3K36me3 modification, a marker for actively transcribed genes (Supplementary Fig. 9b, c). This suggested that SUPT6H, SUPT5H and CTR9 might participate in the same biomolecular condensate to regulate transcription elongation. To confirm the condensation properties of these component CAPs, we performed fixed cell immunofluorescence (IF) with antibodies against SUPT6H, SUPT5H and CTR9 in mESC. We found that all three CAPs can form nuclear puncta in cells (Fig. 4a), which is consistent with a recent study showing the condensation properties of SUPT6H and CTR9 in cells[42]. To determine whether these CAPs coexist in the same puncta, we conducted co-IF analysis and found their high co-localization in nuclei (Fig. 4a, b). And we also conducted fluorescence recovery after photobleaching (FRAP) experiments to check the condensate-like properties of observed nuclear puncta. After photobleaching, SUPT6H, SUPT5H and CTR9 puncta recovered fluorescence rapidly (Fig. 4c–e). To further verify the presence of the associated biomolecular condensate at these CondSig-positive peaks, we conducted CUT&RUN experiments for SUPT6H and CTR9 in wild type, 2,5-HD-treated and 1,6-HD-treated mESC. We observed that CondSig-positive peaks exhibited significantly greater decreases in CUT&RUN signals for both SUPT6H and CTR9 compared to CondSig-negative peaks upon 1,6-HD treatment (Fig. 4f–h, Supplementary Fig. 9d–j), in both comparisons with wild type and 2,5-HD treatment. These results suggested that SUPT6H, SUPT5H and CTR9 can regulate transcription elongation by forming biomolecular condensate.

To further strengthen our validations for identified CondSigs, we next focused on SS18, EP300, and ELL3, which were identified as component CAPs that co-occurred in three CondSigs in mESC (Fig. 2a). While the involvement of SS18 and EP300 in biomolecular condensation has been reported respectively[23,31], ELL3's role remains unexplored, and their co-condensation is not yet known. Through co-immunofluorescence staining, we observed that these three CAPs could form nuclear puncta with high co-localization in nuclei (Supplementary Fig. 10a, b). Furthermore, FRAP experiments confirmed the condensate-like properties of SS18 and ELL3 puncta respectively (Supplementary Fig. 10c, d), complementing the reported validation for EP300[31]. And we also conducted CUT&RUN assays for SS18, EP300, and ELL3 in mESC respectively. The results showed that all three CAPs exhibited a significant reduction at CondSig-positive peaks upon 1,6-HD treatment (Supplementary Fig. 10e–h, Fig. 11a–f).

## Effects of biomolecular condensate on chromatin activities

With the availability of CondSig-positive sites, it is possible to investigate the influence of biomolecular condensates on chromatin activities at a genome-wide scale. Our initial analysis for histone

modifications at CondSig-positive sites revealed a high enrichment of active histone modifications, such as H3K4me3 and H3K27ac, in both mESC and K562 (Fig. 5a), suggesting a close association between biomolecular condensates and chromatin activities. We defined the target genes associated with the CondSig-positive sites (see Methods for details), discovering that these genes showed significantly higher expression levels in both mESC and K562 (Supplementary Fig. 12a, b). Given that transcriptional bursting is a common characteristic of gene expression[43], and it was hypothesized that biomolecular condensation can influence the transcriptional bursting frequencies of target genes[44], we generated single-cell RNA-seq data in wild type, 2,5-HD-treated and 1,6-HD-treated mESC, from which we inferred transcriptome-wide transcriptional bursting kinetics[45]. Among the genes with inferable transcriptional bursting kinetics, those associated with CondSig-positive sites exhibited significantly higher bursting frequencies in the wild type mESC (Fig. 5b). They also displayed a more substantial decrease in transcriptional burst frequencies upon 1,6-HD treatment compared to other genes (Fig. 5c, Supplementary Fig. 12c). After assigning genes associated with CondSig-positive sites to individual CondSig, we ranked the CondSigs in mESC according to the decrease level of transcriptional bursting frequencies upon 1,6-HD treatment. As shown in Fig. 5d, the CondSig containing METTL3, POLR2A, TMPRSS4, CDK8 and EP300 demonstrated the most substantial decrease, suggesting that these CAPs may form biomolecular condensation to enhance the transcriptional bursting frequencies of their target genes. On the contrary, the CondSig containing SUZ12, JARID2, KDM4C, PCGF2, EZH2, RNF2 and CBX7 had the most increase, consistent with their repressive roles in transcription regulation[46]. We also performed transcriptional bursting analysis in K562 and found that CondSig-positive target genes exhibited higher burst frequencies in the wild type K562 (Fig. 5e) and exhibited decreased burst frequency upon 1,6-HD treatment (Fig. 5f, Supplementary Fig. 12d). These results suggested that biomolecular condensation can regulate gene transcription by influencing burst frequency. To rule out the possibility that higher burst frequencies are attributable to the stronger epigenetic modifications at CondSig-positive sites, we compared genes targeted by CondSig-positive / negative sites with the same histone modifications or chromatin accessibility (see Methods for details). We observed that CondSig-positive target genes always showed higher burst frequencies than CondSig-negative targets (Supplementary Fig. 12e, f).

Notably, several histone modification writers, such as EP300 and KMT2D, were included in the components of identified CondSigs. Given the enrichment of their corresponding histone modifications at CondSig-positive sites (Fig. 5a), we hypothesized that these histone modification writers might exhibit stronger catalyzation activities within biomolecular condensates. We classified each histone modification writer's ChIP-seq peaks into CondSig-positive and -negative peaks, and observed significantly stronger corresponding histone modification products at CondSig-positive peaks (Supplementary Fig. 12g, h), suggesting the formation of biomolecular condensation can boost the catalyzation activities of histone modification writers. Active modifications, such as H3K4me3 and H3K27ac, typically display narrow peaks (width <2 kb), while a small proportion also exists as broad peaks (width > 5 kb)[28,47]. The establishment of these broad histone modification domains remains unclear, hence we next investigated whether the involvement of their writer in biomolecular condensation could play a role. We transformed two histone modifications' peaks to domains by merging adjacent peaks not further than 5 kb. Among 1217 H3K4me3 broad peaks in mESC, 63.3% of them overlapped with KMT2D-associated CondSig-positive peaks, while the percentage is only 38.0% for narrow peaks (Fig. 5g). Similar results were observed for the pair of H3K27ac and EP300, not only in mESC, but also in K562 (Fig. 5g, h). These results demonstrated that the involvement of histone modification writers in biomolecular

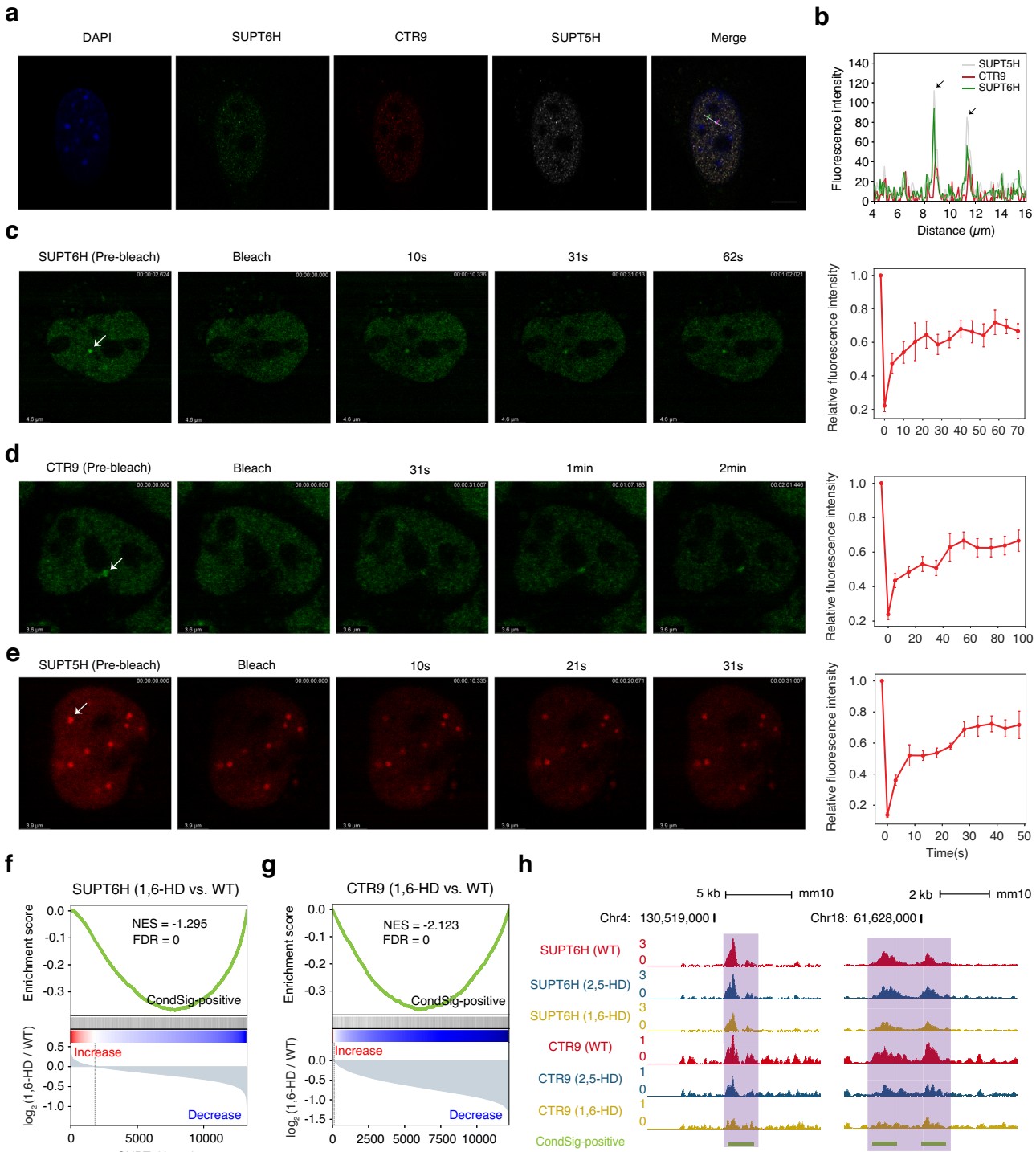

**Fig. 4 | Experimental validation of potential biomolecular condensates.**
**a** Immunofluorescence images of mESC showing that SUPT6H (green) colocalizes with CTR9 (red) and SUPT5H (grey) in puncta. DNA was stained with DAPI (blue). This experimental result was consistent across two independent cell-seeding, fixation, and co-IF staining experiments (each contains three slides). Scale bar: 10 μm. **b** Line scans of the images of a cell co-stained for SUPT6H, CTR9 and SUPT5H, at the position depicted by the white line. The direction is from the green tick to the purple tick, and the two arrows refer to two representative puncta. FRAP experiments for SUPT6H (**c**), CTR9 (**d**) and SUPT5H (**e**). Left, representative images of the FRAP experiment. The white arrow refers to the punctum undergoing bleaching. Right, quantification of FRAP data for puncta of SUPT6H ($n = 6$), CTR9 ($n = 5$) and SUPT5H ($n = 5$). Puncta were photobleached at t = 0 s, and data were plot

as mean ± standard error. Gene set enrichment analysis (GSEA)-like analyses for SUPT6H (**f**) and CTR9 (**g**), with all focus sites (CondSig-positive and -negative peaks of the CAP) were ranked by the log$_2$-transformed fold change in CUT&RUN signals and annotated against the set of CondSig-positive peaks. Fold changes were from 1,6-HD treatment versus wild type, and the pseudo count in fold change calculations was set to 0.1. The GSEA-like analyses were performed with Python package GSEApy (v1.1.2)[68]. The enrichment score profile, the position of CondSig-positive peaks and fold change profiles were shown. The normalized enrichment score (NES) and false discovery rate (FDR) were labeled. **h** UCSC genome browser view of representative CondSig-positive sites. Signals represent RPM and the related loci were shaded in purple. Source data are provided as a Source Data file.

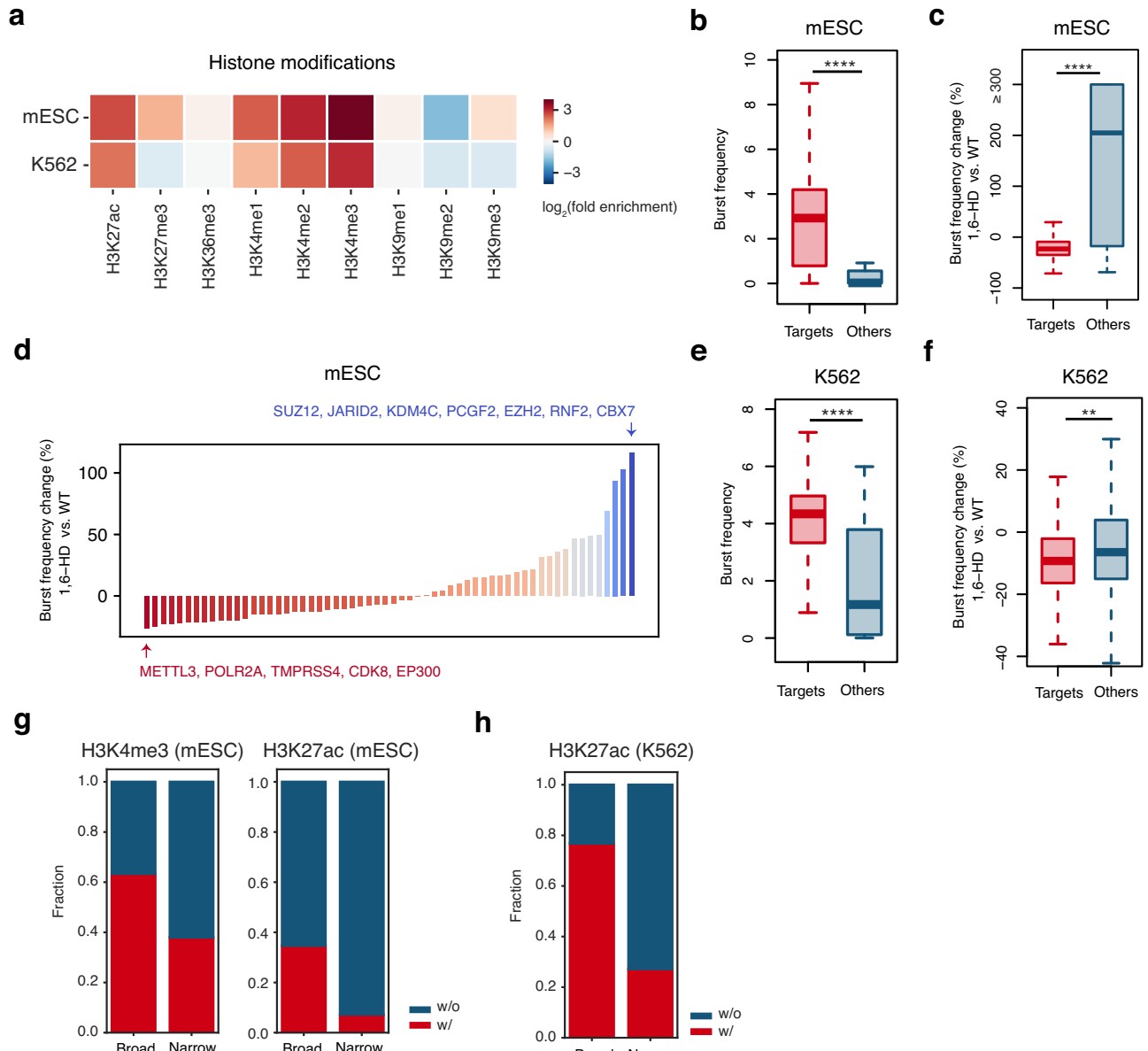

**Fig. 5 | Effect of biomolecular condensates on chromatin activities. a** Heatmaps showing histone modification enrichment at CondSig-positive sites in mESC and K562. The colours represent log$_2$-transformed fold enrichment of histone modification ChIP-seq signals at CondSig-positive sites relative to the genomic background. The genomic background is the average signal of all 1 kb bins in the genome. Public ChIP-seq data for histone modifications were from Cistrome Data Browser[50] and filtered as the previous study described[51]. **b** Box plots comparing burst frequency of genes targeted by all CondSig-positive sites and other genes in mESC. The centre lines mark the median, the box limits indicate the 25th and 75th percentiles, and the whiskers extend to 1.5 × the interquartile range from the 25th and 75th percentiles. Statistical significance between groups was evaluated by a one-sided Welch's *t*-test, **** represents *p*-value < 1 × 10⁻⁴. Sample size used to derive statistics is 1317 for target genes and 134 for other genes. **c** Box plots comparing burst frequency change percentages after 1,6-hexanediol treatment of target genes and other genes in mESC. The centre lines mark the median, the box limits indicate the 25th and 75th percentiles, and the whiskers extend to 1.5 × the interquartile range from the 25th and 75th percentiles. Statistical significance between groups was evaluated by a one-sided Welch's *t*-test, **** represents *p*-value < 1 × 10⁻⁴. Sample size used to derive statistics is 1317 for target genes and

106 for other genes. **d** The bar plots showing burst frequency change percentages of genes targeted by each individual CondSig in mESC. And CondSigs showing the maximum decrease and increase were specially labeled. **e** Box plots comparing burst frequency of genes targeted by all CondSig-positive sites and other genes in K562. The centre lines mark the median, the box limits indicate the 25th and 75th percentiles, and the whiskers extend to 1.5 × the interquartile range from the 25th and 75th percentiles. Statistical significance between groups was evaluated by a one-sided Welch's *t*-test, **** represents *p*-value < 1 × 10⁻⁴. Sample size used to derive statistics is 1279 for target genes and 100 for other genes. **f** Box plots comparing burst frequency change percentages after 1,6-hexanediol treatment of genes targeted by all CondSig-positive sites and other genes in K562. The centre lines mark the median, the box limits indicate the 25th and 75th percentiles, and the whiskers extend to 1.5 × the interquartile range from the 25th and 75th percentiles. Statistical significance between groups was evaluated by a one-sided Welch's *t*-test, ** represents *p*-value < 0.01 and **** represents *p*-value < 1 × 10⁻⁴. Sample size used to derive statistics is 1195 for target genes and 78 for other genes. The stacked bar plots showing fractions of broad H3K4me3 or H3K27ac peaks in mESC (**g**) and broad H3K27ac peaks in K562 (**h**) overlapping with CondSig-positive sites. Source data are provided as a Source Data file.

condensates can alter chromatin activity by catalyzing broad histone modification domains.

## Discussion

The field of biomolecular condensate research associated with chromatin has made substantial advancements in recent years. However, identifying the involvement of a CAP in chromatin-associated biomolecular condensate only scratches the surface of its regulatory roles due to the following inherent limitations. Firstly, biomolecular condensates typically comprise multiple components, each potentially contributing different regulatory roles. Secondly, profiling the genomic binding sites of a CAP involved in a biomolecular condensate does not necessarily distinguish its condensation-associated and non-associated genomic loci in a straightforward manner. Therefore, there is an urgent need for specialized experimental methods or bioinformatic tools to provide a detailed genomic landscape of chromatin-associated biomolecular condensates. A recent study introduced DisP-seq[48], an antibody-independent chemical precipitation assay that maps endogenous DNA-associated disordered proteins at a genomic scale. However, DisP-seq was designed for the broad detection of disordered proteins rather than specifically targeting biomolecular condensates. This could potentially result in both false positives, as not all binding sites of these proteins participate in biomolecular condensates, and false negatives, as disordered protein-guided phase separation is only one mechanism of condensation. Furthermore, DisP-seq cannot identify the exact components present at each locus. In response to these challenges, our study presented CondSigDetector, a computational framework designed to systematically identify CondSigs, i.e., the signatures of condensate-like chromatin-associated protein co-occupancy, and their associated genomic loci. CondSigDetector was designed based on the increasing evidences suggesting chromatin-associated biomolecular condensates are mediated by collaborative interactions of components at specific genomic loci[18]. The key assumption it relies on is that specific collaborations among CAPs on the genome, particularly those involving proteins with high condensation potential (such as known phase-separating proteins and IDR-containing proteins), may act as signatures of chromatin-associated biomolecular condensates. By leveraging the occupancy profiles and condensation-related features of hundreds of CAPs in the same cell type, CondSigDetector can predict the genome-wide loci of biomolecular condensates and the component CAPs of each condensate. Our study both depicted the chromatin properties of the identified CondSigs and experimentally validated the regulatory roles of DDX21, SUPT6H, CTR9 and SUPT5H as components of biomolecular condensates. Our study further delves deeper into the significant effects of chromatin-associated biomolecular condensates on transcriptional bursting and broad active histone modification domains. These findings underscored the critical role that biomolecular condensates play in gene regulation and chromatin activities.

The CondSigs identified in this study provided a comprehensive, global and genome-wide perspective on distinct chromatin-associated biomolecular condensates, paving the way for further exploration of their biological functions and mechanisms. By distinguishing various biomolecular condensates through the unique component CAPs, the CondSigs can not only aid in discovering additional components of known chromatin-associated biomolecular condensates, but also reveal entirely unknown ones. Furthermore, by pinpointing specific genomic loci targeted by biomolecular condensates composed of CAPs, CondSigs provide valuable insights into how dysregulation of condensation may contribute to disease. This, in turn, could facilitate the design of potential therapeutic strategies. To benefit future research in this area, we have made the CondSigs identified in mESC and K562 publicly available online and provided the source code of CondSigDetector on GitHub to enable the detection in other biological systems. However, given the dynamic property of biomolecular

condensates, characterized by their reversible formation and dissolution, it is possible that not all CondSig-positive sites serve as condensate-related sites. To further filter high confidence condensate-related sites from CondSig-positive sites, we required these sites to exhibit elevated concentrations of all component CAPs (i.e., requiring each component CAP to have more than a 1.5-fold increase in signals at condensate-related sites compared to its peaks not associated with CondSigs). By this way, we filtered 15,303 and 16,010 high-confidence condensate-related sites in mESC and K562, which are also publicly available online.

Despite the significant insights provided by our identified CondSigs, there are some limitations to the predictions. One such limitation is the dependence of CondSig detection on accurate occupancy profiles of CAPs. The absence or poor quality of ChIP-seq data could lead to partial or complete omission of biomolecular condensates. For example, we were able to predict a heterochromatin-related condensate consisting of CBX5, TRIM28 and CBX1 in K562, but not in mESC, due to the unavailability of high-quality ChIP-seq data of these CAPs in mESC. However, with the rapid increase of ChIP-seq data, and the implementation of techniques for occupancy map capture, we anticipate improvements in the sensitivity of CondSigs detection. Another limitation is the reliance of CondSig detection on specific collaborations among CAPs, which may result in the loss of widespread collaborations in a global context. In this study, we used a threshold of 1.3 for the z-score normalized occurrence probability of words in topics to determine the component CAPs of CondSigs. Given the lack of a standard number for components in collaborations, the components listed in CondSig might be incomplete or inaccurate, underscoring the need for further in-depth analysis and experiments to verify the predictions. A recent study reported that fixation, a common procedure used in X-ChIP, can have diverse effects on biomolecular condensates in living cells[49]. To assess the potential impact of fixation on our prediction results, we selected several component CAPs with additional available data generated by CUT&RUN, a fixation-free technology, to evaluate the concentration levels in CondSigs. We found that, similar to ChIP-seq signals, most component CAPs showed significantly enriched CUT&RUN signals at CondSig-positive peaks (Supplementary Fig. 13), implying that the fixation effect in the X-ChIP procedure is unlikely to significantly impact prediction accuracy. This potential impact could be further mitigated with the rapid accumulation of more CUT&RUN data for CAPs. In this study, we conducted experiments with 1,6-HD to investigate the impact of condensate disruption on burst frequencies of CondSig target genes, as well as genome-wide occupancy of component CAPs. However, 1,6-HD treatment is harsh and may induce many undesirable effects[39]. Although 2,5-HD treatment can serve as an effective control for 1,6-HD treatment, more stringent experimental approaches are required in future studies on investigating genome-wide effects of condensate disruption.

## Methods

### ChIP-seq data collection and processing
ChIP-seq data of CAPs were collected from Cistrome Data Browser[50] and filtrated using quality control procedures as described in the previous study[51]. In brief, only ChIP-seq data that satisfied at least four out of the five quality control metrics (sequence quality, mapping quality, library complexity, ChIP-enrichment, and signal-to-noise ratio) available in Cistrome Data Browser were kept. When multiple qualified ChIP-seq datasets were available for a given CAP in the same cell type, all qualified ChIP-seq data were sorted based on quality control metrics, and the highest-ranked dataset was selected.

We downloaded ChIP-seq peak files (in BED format) and signal track files (in bigWig format) from Cistrome Data Brower. Although Cistrome Data Browser stored narrow peaks called by MACS2[52] for all CAPs, peak window sizes of distinct CAPs could differ significantly.

Therefore, to obtain accurate occupancy regions for each CAP, especially CAPs with broad peaks, we first called broad peaks from the signal track using "bdgbroadcall" module of MACS2 (v2.1.3) with default parameters and then merged adjacent peaks within 5 kb. For each CAP, if more than 1000 re-called peaks were wider than 5 kb, we replaced the original narrow peaks with re-called broad peaks as the accurate occupancy regions.

## Condensation-related annotation for proteins

Human and mouse proteins with reported LLPS capacity were collected from four databases, DrLLPS[6], LLPSDB[5], PhaSepDB (two versions, v1 and v2)[3] and PhaSePro[4]. DrLLPS collected all proteins that could potentially be involved in LLPS, including scaffolds, regulators and clients. However, we only regarded scaffolds as LLPS proteins since DrLLPS contains too many regulators and clients. To create an annotation of LLPS proteins, we merged all LLPS proteins from different sources. Notably, since the number of collected mouse LLPS proteins (61) was much lower than human LLPS proteins (437), we also considered mouse orthologs of human LLPS proteins as mouse LLPS proteins.

Component proteins of MLOs in human and mouse were collected from DrLLPS and PhaSepDB (v1 and v2). Proteins that were assigned to the same MLO in different sources were merged to form a comprehensive list of component proteins for that MLO. Similar to LLPS proteins, mouse orthologs of human proteins assigned to the same MLO was regarded as component proteins of that MLO in mouse.

Pairwise protein-protein interactions were collected from three databases, BioGRID[53], MINT[54] and IntAct[55], only physical associations were kept.

Intrinsically disordered regions of proteins were predicted by MobiDB-lite (v1.0)[56]. This optimized method uses eight different predictors to derive a consensus, which is then filtered for spurious short predictions in a second step. For each protein, if more than 15.3% of its regions were predicted to be disordered by MobiDB-lite, the protein would be regarded as proteins with intrinsically disordered regions. The threshold of 15.3% corresponds to the 20th percentile of disordered region fractions of known human LLPS proteins.

RNA-binding proteins were predicted by TriPepSVM (v1.0)[57], a method to perform de novo prediction based on short amino acid motifs, with parameters "-posW 1.8 -negW 0.2 -thr 0.28".

## Genome-wide RNA-binding strength

We used genome-wide signals of R-ChIP data, an in vivo R-loop profiling approach using catalytically dead RNase H1[58], to quantify genome-wide RNA-binding strength in K562 cells. Raw sequencing reads from GSE97072[58] were first aligned to human genome build via default --local mode of Bowtie2 (v2.3.5.1)[59]. Low mapping quality reads (mapping quality <30) and duplicates were discarded. Then signal tracks were generated using the "genomecov" command in Bedtools software (v2.28.0), and normalized to reads per million mapped reads (RPM).

## Motif scan

Motif scans were performed using FIMO (v5.0.5)[60] against the JASPAR core 2020 vertebrates database[61] with the following parameters "--max-stored-scores 1000000". Motifs with $p$-values $1 \times 10^{-5}$ were used for the following analysis.

## CondSigDetector workflow

The framework consists of three steps, data processing, co-occupancy signature identification and condensation potential filtration.

In the first step, CondSigDetector first defines the mouse (mm10) or human (hg38) genomic regions as $B_{mm10}$ or $B_{hg38}$ as a sequence of 1 kb consecutive bins $B = \{b_1, b_2, \cdots, b_n\}$, where each $b_i$ represents $i$-th 1 kb bin and $n$ is the total number of 1 kb bins in the genome. And it

defines the set of CAPs as $C = \{c_1, c_2, \cdots, c_m\}$, where each $c_j$ represents $j$-th CAP and $m$ is the total number of CAPs. Then it generates an occupancy matrix $O$ with dimension $n \times m$, each element $O_{i,j}$ of the matrix $O$ represents the occupancy event of $j$-th CAP at $i$-th bin, which is defined as:

$$O_{i,j} = \begin{cases} 1, & \text{if CAP } j \text{ has a peak within bin } i \\ 0, & \text{otherwise} \end{cases} \tag{1}$$

CondSigDetector further apply filters to the occupancy matrix $O$ to refine the data. It excludes CAPs with fewer than 500 occupancy events to eliminate the effect of low-quality ChIP-seq data. And bins with too many occupancy events (occupied by more than 90% of CAPs) are removed to avoid sequencing bias. Additionally, bins in ENCODE Blacklist genomic regions are also removed.

Identifying co-occupancy signatures from the entire occupancy matrix $O$ is a complicated task and can result in the loss of low-frequency signatures in the local context. To address this issue, CondSigDetector iteratively segments the entire occupancy matrix into sub-matrices, each sub-matrix contains high-frequency co-occupancy events associated with the given CAP in each iteration. The segmentation of each iteration includes two aspects, identifying the segment of CAPs showing specific co-occupancy with the given CAP and identifying the segment of bins showing high co-occupancy events of these CAPs. During each segmentation iteration, CondSigDetector selects a focus CAP $c_f$, and identifies other CAPs $C_{segment} \subset C$ that are highly co-occupied with $c_f$. The identification of $C_{segment}$ is calculated as follows: for each CAP $c_j \in C$, CondSigDetector uses its occupancy events $[O_{1j}, O_{2j}, ..., O_{nj}]^T$ to classify occupancy events of $c_f$ $[O_{1f}, O_{2f}, ..., O_{nf}]^T$ and calculates a $F_1$ score as a measure of co-occupancy level with $c_f$, denoted as $\beta_j$. The top $q$ CAPs ranked by $\beta_j$ are kept as $C_{segment}$, where $q = 50$ by default. After that, CondSigDetector further selects bins $B_{segment} \subset B$ that are occupied frequently by CAPs $C_{segment}$. The selection of $B_{segment}$ is calculated as follows: for each bin $b_i \in B$, CondSigDetector calculates an occupancy score $\delta_i$ to evaluate the occupancy level of the CAPs $C_{segment}$ as:

$$\delta_i = \sum_{j=1}^{q} \gamma_j O_{ij} \tag{2}$$

Where $\gamma_j$ denotes $z$-score-normalized $\beta_j$. Only $p$ bins with $\delta_i > 0$ are kept as $B_{segment}$. Sub-matrix in each iteration $O_{segment}$ is defined as:

$$O_{segment} = \left[ O_{ij} \right]_{b_i \in B_{segment}, c_j \in C_{segment}} \tag{3}$$

In the second step, each sub-matrix $O_{segment}$ is classified into promoter and non-promoter contexts. Promoters are defined as upstream 3 kb to downstream 3 kb of transcription start sites. CondSigDetector builds a biterm topic model[22] for each sub-matrix to learn specific collaborative pattern of CAPs, termed co-occupancy signatures. The topic model is a well-common used machine learning model for discovering latent topics in a particular set of documents, and it assumes that each document can be described as a mixture of a small number of topics, where a topic is a distribution of words. CondSigDetector lets $D = \{d_1, d_2, \cdots, d_p\}$ be a collection of "documents", where each document $d_i$ corresponds to occupancy events at $i$-th bin $O_i$, and lets $W = \{w_1, w_2, \cdots, w_q\}$ be the "vocabulary", where each word $w_j$ corresponds to a CAP $c_j$. Then the learned latent topics across "documents" can be regarded as specific collaborative pattern of CAPs across genome.

The biterm topic model is a type of probabilistic topic model designed to find topics in collection of short texts, and the goal is to

learn: (1) $\theta_{i,t}$, which is the probability of topic $t$ occurring in document $d_i$; (2) $\Phi_{t,j}$, which is the probability of word $w_j$ belonging to topic $t$. We implemented the topic model in CondSigDetector using source code from the previous study[22]. Finally, the biterm topic model generates two probability distributions, matrix $G_{k \times q}$ representing occurrence probability of $q$ words across $k$ topics and matrix $G_{p \times k}$ representing occurrence probability of $k$ topics across $p$ documents.

The topic number, $k$, is a crucial parameter in topic modeling, as it affects the topic distribution. CondSigDetector empirically learns 2 ~ 10 topics for each context and then applies an automatic strategy to select the optimal topic number as described in the previous study[62]. The selection principle was based on the idea that the optimal topic number should distinguish between documents with different topics as much as possible. Hence an optimal topic number should match the following two criteria:

- The occurrence probability of each topic in different documents should be as different as possible, which is measured by the specificity score ($SS_k$) calculated for all topics under a certain topic number $k$.

$$SS_k = \log\left(\frac{1}{k}\sum_{j=1}^{k}\frac{\sigma_j}{\mu_j^2}\right) \qquad (4)$$

where $\sigma_j$ and $\mu_j$ are the variance and mean, respectively, of the $j$-th column of $G_{p \times k}$. A higher specificity score indicates a better-selected topic number.

- The fewer topics that occur in each document, the better. Such a measurement was defined as a purity score ($PS_k$) for all topics under a certain topic number $k$.

$$PS_k = \log\left(\frac{1}{p}\sum_{i=1}^{p}\sigma_i\right) \qquad (5)$$

where $\sigma_i$ is the variance of $i$-th row of $G_{p \times k}$. The larger the purity score, the better the selected topic number.

Finally, we defined the combination score ($CS_k$), which is a weighted average of the specificity score and purity score. The combination score ($CS_k$) is calculated as

$$CS_k = \alpha SS_k + (1-\alpha)PS_k \qquad (6)$$

where $\alpha$ is calculated as

$$\alpha = \frac{PS_k}{SS_k + PS_k} \qquad (7)$$

We selected the optimal topic number $k$ from 2 - 10 which have the highest combination score.

After the selection of optimal topic number $k$, CondSigDetector interpretated learned topics to co-occupancy signatures. We determined component CAPs of each co-occupancy signature based on matrix $G_{k \times q}$ representing $q$ CAPs' occurrence probability in $k$ co-occupancy signatures. For each signature $t$, a CAP $c_j$ was considered as a component if $Z(G_{t,j})>\lambda$, where $Z$ is the $z$-score normalization function and $\lambda$ is the threshold set to 1.3 by default. And a 1 kb bins $b_i$ was defined as signature-positive sites if it is occupied by more than 80% of components CAPs. By this way, we generated component CAPs $C_{pos}$ and signature-positive sites $B_{pos}$. Co-occupancy signatures with fewer than 3 components and fewer than 200 signature-positive sites are discarded.

In the third step, CondSigDetector screens out CondSigs from all co-occupancy signatures based on the condensation potential of each signature. To evaluate the condensation potential of each

signature, we quantify associations between condensation-related features and signature presence at genome-wide bins for each signature. Intuitively, the higher condensation-related feature values of occupancy events at signature-positive bins, the higher condensation potential of the signature. We conduct Receiver Operating Characteristic (ROC) curve analysis to compare the distribution of condensation-related feature values at signature-positive versus signature-negative bins and measure the enrichment of condensation-related features at signature-positive bins. In ROC analysis, the positive set are positive bins for the given signature and the negative set are negative bins. Signature-positive bins have been defined in the above step, and signature-negative bins $B_{neg}$ are defined using the following two criteria:

- Comparability with the signature-positive bins. As the signature-positive bins are occupied by at least 80% of $C_{pos}$, so we required that signature-negative bins are occupied by at least $h$ CAPs, with $h = 0.8 \times |C_{pos}|$;
- Differentiation from the signature-positive bins. We required that signature-negative bins are absence of co-occupancy events of $C_{pos}$; specifically, count of occupied $C_{pos} < 2$.

For each signature, six condensation-related features are calculated according to co-occupancy events of $C_{segment}$ at signature-positive bins $B_{pos}$ and signature-negative bins $B_{neg}$:

- $F_{LLPS} = \dfrac{Number\ of\ occupied\ CAPs\ with\ reported\ LLPS\ capacity}{Total\ number\ of\ occupied\ CAPs}$

- $F_{MLO} = \dfrac{Number\ of\ occupied\ CAPs\ co-occuring\ in\ the\ same\ MLO}{Total\ number\ of\ occupied\ CAPs}$

- $F_{IDR} = \dfrac{Number\ of\ occupied\ CAPs\ with\ predicted\ IDRs}{Total\ number\ of\ occupied\ CAPs}$

- $F_{PPI} = \dfrac{Number\ of\ occupied\ CAP\ pairs\ with\ protein-protein\ interactions}{Total\ number\ of\ CAP\ pairs}$

- $F_{RBP} = \dfrac{Number\ of\ occupied\ CAPs\ predicted\ as\ RBPs}{Total\ number\ of\ occupied\ CAPs}$

- $S_{RBS} = RNA\ binding\ strength\ at\ the\ bin$

A signature is identified as a CondSig if at least three out of six condensation-related features exhibit a positive correlation with the presence of the signature, which is measured by the Area Under the ROC Curve (AUROC). The criteria for this identification are an AUROC greater than 0.6 for individual features and a mean AUROC greater than 0.65 for the top three features.

In the final stage, CondSigs within the same cell type are pooled, and any duplicated CondSigs are discarded. The redundancy of two CondSigs is measured based on the extent of overlap among their top five components, ranked by their probability of occurrence within each CondSig. We computed the Jaccard index for each pair of CondSigs. If the Jaccard index suggests a high redundancy (a value greater than 0.25), we then compare the mean AUROC of the two CondSigs and discard the one with low mean AUROC.

## Comparison of BTM and HDP

We built HDP and BTM models on the entire occupancy matrix separately, and compared the quality of learned topics. HDP determines the topic number automatically while BTM asks for a given topic number.

So we first built an HDP model and generated $k$ topics, then we built a BTM model to generate topics with the given topic number $k$. The quality of each learned topic was evaluated by the coherence score of the top five words, a common quality evaluation metric in topic model[22,63]. HDP modeling was implemented by using a Python package "tomotopy".

## Clustering of component CAPs

We performed a $k$-means clustering for component CAPs in mESC or K562 according to their potentials for self-assembly (PS-Self) or interaction with partners (PS-Part) to undergo phase separation. A recent study employed two machine-learning models, SaPS and PdPS model, to estimate proteins' potentials and provided SaPS and PdPS ranking scores (ranging from 0 to 1) for the human and mouse proteome. We utilized the SaPS and PdPS ranking scores of component CAPs in mESC or K562 to carry out $k$-means clustering. In the clustering, the number of clusters was set as 4, and the initial cluster centroids were set as (0.8, 0.8), (0.8, 0.4), (0.4, 0.8), (0.4, 0.4), which corresponds to four clusters: "both Self and Part", "Self-only", "Part-only", and "none", respectively.

## Annotation for charged amino acid blocks

We calculated NCPR (net charge per residue) employing a 10-residue sliding window with a step size of 1. This calculation factored in both positively charged amino acids (R, K and H) and negatively charged amino acids (D and E). Windows with NCPR greater than 0.5 or less than -0.5 were defined as charged amino acid blocks, and overlapping blocks were merged.

## Identification of CondSig-positive and -negative peaks and domains

CondSigDetector identified CondSigs and assigned genome-wide 1 kb bins to each CondSig. In the chromatin properties and disruption effect evaluation of CondSigs, we defined CondSig-positive and -negative peaks/domains for each component CAP to examine its chromatin properties and disruption effect within these identified CondSigs. To determine CondSig-positive and -negative peaks for the given CAP, we classified its ChIP-seq peaks into CondSig-positive or -negative peaks based on whether they overlapped with sites where the CAP was identified as a component of any CondSigs. To determine CondSig-positive and -negative domains, we transformed peaks of the given CAP into domains by merging adjacent peaks not further than $n$ kb. For component CAPs using narrow peaks as accurate occupancy regions in ChIP-seq data processing procedure as mentioned above, we set $n = 5$, and for component CAPs using broad peaks as accurate occupancy regions, we set $n = 10$. Then domains of each component CAP were classified into CondSig-positive domains and -negative domains based on overlapping with CondSig-positive peaks.

To ensure a fairer comparison, two additional criteria were further applied to refine the identification of CondSig-positive and -negative peaks, ensuring they are matched in terms of chromatin accessibility or the number of co-occupied CAPs. Both refined CondSig-positive and -negative peaks were required to overlap with ATAC-seq peaks or have the occupancy events of more than 10 CAPs. These refined CondSig-positive and -negative peaks were then transformed into refined CondSig-positive and -negative domains in the same way.

## 3D chromatin contact analysis

Public Micro-C data in mESC, ChIA-PET data against SMC1 in mESC, and ChIA-PET data against RNA Pol II in K562 were used in this study. Micro-C contact matrices from 2.6 billion reads were downloaded from GSE130275[33], and boundary strength for 400 bp resolution calculated by Cooltools[64] was used for the following analysis. SMC1 ChIA-PET data in mESC were downloaded from GSE57911[36] and processed with ChIA-PET2[65]. RNA Pol II ChIA-PET loops were directly downloaded from ENCSR880DSH[37].

## Definition for target genes of CondSig-positive sites

In defining target genes of CondSigs, positive sites of all identified CondSigs were merged into a total set of CondSig-positive sites. Genes whose promoter overlaps with the CondSig-positive sites, or which have long-range chromatin contacts with those sites, were defined as target genes. These long-range chromatin contacts were determined using ChIA-PET data from the corresponding cell type. In this study, SMC1 ChIA-PET data in mESC and RNA Pol II ChIA-PET data in K562 were used.

To rule out the possibility that higher burst frequencies are attributable to the stronger epigenetic modifications at CondSig-positive sites, target genes of all CondSig-positive and all CondSig-negative sties with the same histone modifications or chromatin accessibility were also defined. Here, all CondSig-negative sties (i.e., the total set of CondSig-negative sites) were specified as any 1 kb genomic bins occupied by at least two CAPs and not identified as CondSig-positive. Both CondSig-positive and -negative sites were first intersected with ChIP-seq peaks of H3K4me3 or H3K27ac, or ATAC-seq peaks, and their target genes were then defined in the same manner mentioned above.

## Cell culture

Mouse embryonic stem cells (mESC), C57BL/6 strain, were purchased from ATCC (SCRC-1002) and cultured on a feeder layer of mitomycin C (Stemcell, 73272) treated mouse embryonic fibroblast (MEF) in tissue culture flask coated with 0.1% gelatin. The cells were grown in complete mESC medium, which was composed of EmbryoMax DMEM (Millipore, SLM-220-B), 15% (v/v) fetal bovine serum (Hyclone, SH30070.03), 0.1 mM nonessential amino acids (Millipore, TMS-001-C), 1% (v/v) nucleoside (Millipore, ES-008-D), 2 mM L-glutamine (Millipore, TMS-002-C), 0.1 mM β-mercaptoethanol (Millipore, ES-007-E), and 1000 U/mL recombinant LIF (Millipore, ESG1107).

## Cell treatment

1,6-hexanediol (Sigma, 240117) was dissolved in a complete mESC medium at a concentration of 15% (w/v) to make a storage solution, similarly, 2,5-hexanediol (Sigma, H11904) was prepared at a 15% (v/v) concentration in the same medium. mESC were detached using trypsin, pelleted by centrifuging, and then resuspended in a complete mESC medium. The resuspended cells were transferred into a gelatin-coated flask and cultured in a 37 °C incubator for 1 hr to remove the feeder cells. The supernatant cells were collected and washed twice with PBS. After cell resuspending with medium, either 1,6-hexanediol or 2,5-hexanediol storage buffer was added at a final concentration of 1.5%. The dishes were put into the incubator immediately for 30 min, and treated cells were immediately used for CUT&RUN assay.

## CUT&RUN

The CUT&RUN assay was conducted on 0.2 million cells per sample, utilizing the Hyperactive pG-MNase CUT&RUN assay kit (Vazyme, HD102) with slight modifications to the manufacturer's protocol. Briefly, cells were harvested and incubated for 10 min at room temperature with Concanavalin A-coated magnetic beads, which had been activated prior to use. Following this, the ConA beads bound cells were collected using a magnet and resuspended in 100 μl of antibody buffer containing either 2 μl of DDX21 (Proteintech, 10528-1-AP, lot # 00088037), 4 μl of CTR9 (Bethyl Laboratories, A301-395A, lot # 4), 4 μl of SUPT6H (Novus Biologicals, NB100-2582, lot # 2 A), 1.5 μl of SS18 (Cell Signaling Technology, 21792 (D6I4Z), lot # 1), 1.5 μl of EP300 (Santa Cruz, sc-48343 (F-4), lot # A1323), or 0.5 μl of ELL3 (generously gifted by Prof. Chengqi Lin, Southeast University, China) primary

antibody respectively. The samples were then incubated at 4 °C overnight on rotator. The next day, cells were washed twice with Dig-wash buffer and resuspended in 100 μl of a premixed pG-MNase Enzyme solution before incubation at 4 °C for 1 hr with rotation. Following this, the cells were washed twice with Dig-wash buffer and resuspended in 100 μl of premixed CaCl₂ solution, then incubated for 2 h on ice. Following the stop of the reaction, the cut chromatin was released from cells by incubation at 37 °C for 30 min in the absence of agitation. After centrifuging at 13,400 g for 5 min, the supernatant was collected, and DNA was purified using FastPure gDNA mini columns. The libraries were prepared using NEBNext Ultra II DNA library prep kit (NEB, E7645) with modified amplification condition as 98 °C for 30 s, 15 cycles of 98 °C for 10 s and 65 °C for 17 s, and final extension at 65 °C for 2 min and hold at 4 °C.

### Single-cell RNA-seq

Single-cell RNA sequencing (scRNA-seq) libraries were prepared using 6000 mES cells, either in a wild type state or treatment with 1,6-hexanediol at 1.5% or 2,5-hexanediol at 1.5% for 30 min, and K562 cells (National Collection of Authenticated Cell Cultures, TCHu191), either in wild type or treatment with 1,6-hexanediol at 10% for 20 min. The libraries were created using the Chromium Single Cell 3' Library and Gel Bead Kit V3.1 (10x Genomics, Catalog No. PN1000268) to create single-cell gel beads in emulsion (GEM). Following preparation, the libraries were sequenced using the Illumina Novaseq 6000 platform in a 150 bp paired-end mode.

### Immunofluorescence staining

CTR9 antibody was labeled with Mix-n-Stain CF488 Antibody labeling kit (Sigma, MX488AS20), while SUPT6H and ELL3 (Sigma, HPA028938) antibodies were labeled using Mix-n-Stain CF568 Antibody labeling kit (Sigma, MX568S20) according to the manufacturer's instruction. For co-immunofluorescence study of SUPT6H/CTR9/ /SUPT5H, mESC were grown as mentioned above on pre-coated coverslips and fixed with 4% paraformaldehyde solution (Beyotime, P0099) at room temperature for 10 min. permeabilization was performed using 0.5% Triton X-100 (Sigma-Aldrich, 93443) in PBS for 10 min. Cells were blocked with IF blocking solution (Beyotime, P0102) for 1 h at RT, and subsequently incubated with a 1:100 diluted SUPT5H primary antibody (Santa Cruz, 133217 (D3), lot # G1217) in QuickBlock dilution buffer (Beyotime, P0262) at 4 °C overnight. Following three washes, cells were incubated with Alexa Fluor 594 goat anti-rabbit secondary antibody (ThermoFisher, A11037) at a concentration of 1: 1000 in PBST for 1 h at RT. After three additional washes with PBST, cells were labeled with both CF488-conjugated CTR9 (1:250 diluted) and CF568-conjugated SUPT6H (1:200 diluted) antibodies at RT for 2 h. After three washes with PBST, the coverslips were mounted onto glass slides using Vectashield medium with DAPI (Vector Laboratories, H-1200) and sealed with nail polish. Similarly, For the co-IF experiment of SS18/EP300/ELL3, blocked mESC were incubated with 1:400 diluted SS18 and 1:200 diluted EP300 (Santa Cruz, 32244 (NM11), lot # H1921) primary antibodies, followed by incubation with Alexa Fluor 488 goat anti-rabbit (ThermoFisher, A11008) and Alexa Fluor 594 goat anti-mouse (ThermoFisher, A11032) secondary antibodies at a concentration of 1: 1000 for 1 h at RT, then labeled with 1:200 diluted CF568-conjugated ELL3 antibody for 2 h. Images were acquired using a Zeiss LSM 710 confocal microscope with 100 × oil objective and ZEN acquisition software.

### Fluorescence recovery after photobleaching (FRAP)

FRAP assay was conducted using the FRAP module of the Leica SP8 confocal microscopy system. The CTR9 and SUPT6H endogenously tagged with EGFP in mESC was bleached using a 488 nm laser beam. The mScalet-SUPT5H overexpressed in mESC was bleached using a 561 nm laser beam. Similarly, mESC overexpressed with SS18-EGFP

and EGFP-ELL3 were bleached using a 488 nm laser beam. Bleaching targeted a specific circular region of interest (ROI) using 100% laser power and time-lapse images were collected. Fluorescence intensity was measured using Fiji software, with background intensity subtracted and values normalized to pre-bleaching time points.

### CUT&RUN, single-cell RNA-seq data processing

CUT&RUN reads were first processed using TrimGalore (v0.6.0) to trim adaptor and low-quality reads. Trimmed reads were then aligned to the mouse genome build mm10 or human genome build hg38 using Bowtie2 (v2.3.5.1)[59] with parameters "--no-mixed --no-discordant --no-unal". Low mapping quality reads (mapping quality <30) and duplicates were discarded. Then biological replicates that passed quality control were pooled together. For the same CAP, the reads in each condition was down-sampled to the same number. This number was determined by the minimum reads of the CAP across different conditions: 40 million for DDX21 and SS18, and 50 million for the others. CUT&RUN peaks were called by MACS2 (v2.1.3)[52]. Signal tracks were generated using the "genomecov" command in Bedtools software (v2.28.0), and normalized to reads per million mapped reads (RPM). Single-cell RNA-seq data (10x Genomics) were processed with DrSeq2 (v2.2.0)[66] and transcriptome-wide transcriptional burst kinetics were inferred using the model from the previous study[45].

### Statistics and reproducibility

Statistical analysis was performed using Python, and the statistical details are shown in the figure legends and Source Data. No statistical method was used to predetermine sample size. No data were excluded from the analyses. This study did not include complex treatment conditions, all cells were randomly assigned to each group for imaging and sequencing. All samples were prepared blinded.

### Reporting summary

Further information on research design is available in the Nature Portfolio Reporting Summary linked to this article.

## Data availability

All the CUT&RUN and scRNA-seq data generated in this study have been deposited in Genome Sequence Archive (https://ngdc.cncb.ac.cn/gsa/) under accession code CRA011710 and HRA005013. All predicted CondSigs, the associated CondSig-positive sites, and high-confidence condensate-related sites generated in this study are available at CondSigDB (https://compbio-zhanglab.org/CondSigDB/index.html). Source data are provided with this paper.

## Code availability

The computational framework and statistical analysis were made based on shell, Python and R codes. A command-line tool was developed for the implementation of CondSigDetector, main source codes are available at the GitHub repository (https://github.com/TongjiZhanglab/CondSig), which are also deposited at Zenodo[67] (https://doi.org/10.5281/zenodo.12526192).

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

## Acknowledgements

We would like to thank Shuang Hou and Yanhong Xiong for their assistance in statistical analysis, Mengtan Xing for comments on the experimental design, and Chengqi Lin for his technical assistance. We also thank the staff members of the Integrated Laser Microscopy System at the National Facility for Protein Science in Shanghai (NFPS), Shanghai Advanced Research Institute, Chinese Academy of Sciences, China for sample preparation, data collection and analysis. This work was supported by the National Natural Science Foundation of China (32030022 (Y.Z.), 32325012 (Y.Z.), 31970642 (Y.Z.)), the National Key Research and Development Program of China (2021YFA1302500 (Y.Z.)), the Science and Technology Commission of Shanghai Municipality (23JS1401200 (Y.Z.)), the Postdoctoral Innovation Talents Support Program (BX20230265 (Z.Y.)), the China Postdoctoral Science Foundation (2022M722423 (Z.Y.)), the Shanghai Science and Technology Rising Star (22QA1412200 (G.Z.)), the Fundamental Research Funds for the Central Universities (22120220592 (Q.W.)) and the GHfund C (202302033256 (Z.Y.)).

## Author contributions

Y.Z. conceived and designed the research. Z.Y. developed the computational framework and performed computational analysis. Q.W. and Q.Z. performed experiments with the help of Y.T., G.Y., G.Z. and J.Z., Z.Y, Q. W., Q. Z., G. Z. and Y. Z. wrote the manuscript.

## Competing interests

The authors declare no competing interests.
