## [Peer Review File · Nature Communications]

Decoding the genomic landscape of chromatin-associated biomolecular condensatesReviewers' Comments:

Reviewer #1:

None

Reviewer #2:

Remarks to the Author:

This is an interesting manuscript by Yu et. al in which a new computational approach is introduced to identify proteins that are part of chromatin-associated biomolecular condensates. CondSigDetector integrates ChIP-seq data and condensation-related features to identify proteins that co-occur in condensates at specific genomic loci. Predictions are validated by using omics data sources, published condensation data and new experiments.

Overall, I find the approach compelling. However, I find some of the validation not so convincing or even circular. These key issues need to be addressed:

1. The method section is very poorly written. There is often a word salad that is hard to digest. The authors should express their approach in formulas with well-defined variables.
2. There appears to be circularity in some of the validations. The authors use co-occurrence in MLOs as a feature in the prediction pipeline (line 600 in the text). Then, they use membership to known chromatin-associated condensates (line 166) for validation. MLO membership should be excluded as a feature (or at least chromatin, transcription and RNA-processing related MLOs). Otherwise, the enrichment for chromatin-associated condensate members is not surprising. Similarly, the fraction of CAPs with predicted IDRs is used as a feature and DisP-seq used for validation. Again, there may be some circularity here.
3. The authors use proteins that are already known to phase separate for validation (DDX21, SUPT6H and CTR9). It would be much more compelling to use proteins that are not yet known to be part of condensates and/or phase separate to validate their predictions.
4. Related to the previous point, 1,6-HD treatment is very harsh and controversial. It can have many undesirable effects. Introducing mutations that abrogate phase separation for well characterized proteins like MED1 (line 332) and testing effects on transcriptional bursting would be much more compelling.
5. The authors do not confirm the liquid-like properties of the puncta they find (FRAP). The puncta could be all kinds of assemblies.
6. Fig 5b and c, the pronounced effect on busting may be related to the histone modifications that are associated with the CondSig-positive sites. A proper control would be targets of CondSig-negative sites with the same histone modifications.
7. Fig 2d, only IDRs of the proteome should be used as control, not the entire proteome.
8. The authors classify CAPs as scaffolds or clients. I am not sure their classification based on PS-self and PS-part is correct. PS-self is for phase separation driven by homotypic interactions and PS-part for heterotypic interactions. However, the involved proteins may be considered as scaffolds in both cases.
9. Stats: Lots of trends are assessed statistically. Are p values corrected for multiple testing?

Reviewer #3:

Remarks to the Author:

The manuscript from Yu and coworkers describes a new computational framework to predict signatures of proteins bound to the chromatin in a condensate-like manner. They use the resulting CondSigDetector framework to predict known and previously unknown putative chromatin-associated condensates. While the approach could potentially be interesting, several methodological and technical concerns have to be addressed to show the validity of the CondSigDetector as outlined below.

- The authors state that DNA binding co-occupancy could not be explained by DNA motifs or chromatin accessibility. However, the presence of histone modifications or rather interactions with

others proteins may provide some information here. The authors should try to assess this, before claiming that the most-likely alternative explanation lies in condensate formation (line 95-98).

- In general, how CondSigDetector works and what the underlying assumptions are, is not well explained and should be improved in a revised manuscript.
- The authors identify almost 50k sites that are supposedly involving CondSigs. This could easily correspond to half of all CREs in a given cell type. Is that more than expected? Are all sites then regulated by formation of condensate-like structures?
- The authors distribute regions into Condsig-positive and -negative sites, but how do these sites relate to each other as no information is given as to how the CondSig-negative regions are chosen? Further, if the positive regions have higher accessibility and thus a higher number of TFs bound, how fair is it to compare this to negative regions with lower accessibility? It would be better to have regions that are matched in terms of genomic distribution, size and accessibility and then to compare the performance of the CondSigDetector algorithm on this.
- There is no evaluation on the CondSigDetector in term of accuracy on positive control sites for example (e.g. sites that are impacted in terms of accessibility / H3K27ac upon BRD4 inhibition).
- How does the CondSigDetector perform when specific TFs, or specific TF classes, are omitted from the training of the algorithm?
- In line 189-190, the authors state that a certain percentage of CAPs identified as CondSigs are scaffolds or scaffold-clients, but this may indirectly be art of the CondSigDetector categorization as related features such as co-occurrence in MLOs were used in the training. Same holds true for the DISP-seq, as CAPs with a higher percentage of IDRs will be higher valued in the CondSigDetector algorithm, thus this is a self-conformation in a way?
- In general, the authors make very strong claims while they only provide very limited data to support these. And thus some tuning down statements would highly benefit the credibility of the presented manuscript.
- The authors do only one experimental assay, using 1,6-hexanediol (1,6-HD). But this assay is very harsh, and proper controls should be included, for example 2,5-HD. And a lot more data should be shown besides a fold change and one genomic region. That makes any reader doubt how big the changes actually are. Where is an overview of total numbers of DE peaks, genome-wide overview, more example loci?
- The authors frequently compare fold changes at CondSig-positive and negative regions, and claim higher fold changes as the positive regions. But these had for example higher signal to start with, so you would also get higher fold changes. So again, how fair is this comparison?
- The authors talk about higher bursting at Condsig positive sites, but how do these relate to the CondSig negative sites in terms of genomic location, distance to genes, etc?
- Figure 5, what is genomic background?

Reviewer #2 (Remarks to the Author):

This is an interesting manuscript by Yu et. al in which a new computational approach is introduced to identify proteins that are part of chromatin-associated biomolecular condensates. CondSigDetector integrates ChIP-seq data and condensation-related features to identify proteins that co-occur in condensates at specific genomic loci. Predictions are validated by using omics data sources, published condensation data and new experiments.

Overall, I find the approach compelling. However, I find some of the validation not so convincing or even circular. These key issues need to be addressed:

1. The method section is very poorly written. There is often a word salad that is hard to digest. The authors should express their approach in formulas with well-defined variables.

We appreciate your guidance in improving the written quality of our method section. In the revised method section, we added a detailed description along with explicit formulas and well-defined variables to clarify our method. We believe this adjustment makes our method section more straightforward and comprehensible.

2. There appears to be circularity in some of the validations. The authors use co-occurrence in MLOs as a feature in the prediction pipeline (line 600 in the text). Then, they use membership to known chromatin-associated condensates (line 166) for validation. MLO membership should be excluded as a feature (or at least chromatin, transcription and RNA-processing related MLOs). Otherwise, the enrichment for chromatin-associated condensate members is not surprising. Similarly, the fraction of CAPs with predicted IDRs is used as a feature and DisP-seq used for validation. Again, there may be some circularity here.

We appreciate your comments on our validation approach. In the revised manuscript, we removed all MLO memberships used in the training set from the collected list of chromatin-associated condensates. Using the new list, we updated the percentages of CondSigs with known components. As shown in the updated Fig. 2b, 88.0% / 91.7% of CondSigs identified in mESC and 36.0% / 40.9% of CondSigs identified in K562 contain known components.

For the DisP-seq issue, we agreed that the binding sites of IDR-containing proteins (IDPs) are proposed to have higher DisP-seq signals. We further compared the DisP-seq signals at CondSig-positive sites with a control group of sites occupied by any IDPs (termed as IDP-positive sites), and we found that CondSig-positive sites exhibited higher DisP-seq signals than IDP-positive sites (Fig. R1). Nevertheless, to eliminate the possibility of circularity, we removed this part from our revised manuscript.

Fig. R1. DisP-seq signals around CondSig-positive sites and IDP-positive sites.

To enhance the validation of identified CondSigs, we utilized PSPire (Hou *et al.* Nat Commun, 2024), a recently developed phase-separating proteins (PSPs) predictor that incorporates both residue-level and structure-level features. After removing known PSPs from the component CAPs of identified CondSigs, the remaining component CAPs still exhibited higher PSPire scores than all IDPs predicted by MobiDB-lite or the entire proteome, indicating the high phase separation capacities of identified component CAPs (Fig. R2). We added the result as updated Fig. 2e.

Fig. R2 (*i.e.*, Fig. 2e). PSPire scores of component CAPs (with or without known PSPs), all IDPs and the entire proteome. Statistical significance was evaluated by a two-sided Welch's *t*-test.

3. The authors use proteins that are already known to phase separate for validation (DDX21, SUPT6H and CTR9). It would be much more compelling to use proteins that are not yet known to be part of condensates and/or phase separate to validate their predictions.

Thank you for your valuable comments. It should be clarified that the study on the condensation properties of SUPT6H and CTR9 (Lyons *et al.* Cell, 2023) was published shortly after we finished most of our experimental validation on DDX21, SUPT6H and CTR9.

According to your comment, we performed more experimental validations during the revision. We focused on SS18, EP300, and ELL3, which were identified as component CAPs that co-occurred in three CondSigs in mESC. While the involvement of SS18 and EP300 in biomolecular condensation has been reported, ELL3's role remains unexplored, and their co-condensation is not yet known. Through co-immunofluorescence staining, we observed that these three CAPs could form nuclear puncta with high co-localization in nuclei (Fig. R3a,b, *i.e.*, updated Fig. S9a,b). Furthermore, FRAP experiments confirmed the liquid-like property of SS18 and ELL3 puncta respectively (Fig. R3c,d, *i.e.*, updated Fig. S9c,d), complementing the reported validation for EP300 (Ma *et al.*, Mol Cell, 2021). To further verify the presence of the associated biomolecular condensate at these CondSig-positive sites, we conducted CUT&RUN assays for SS18, EP300, and ELL3 in mESC respectively. The results showed that all three CAPs exhibited a significant reduction at their CondSig-positive peaks upon 1,6-HD treatment compared to CondSig-negative peaks (Fig. R3e-h, *i.e.*, updated Fig. S9e-h).

Fig. R3 (*i.e.*, Fig. S9). Additional experimental validations of identified CondSigs.

a. Immunofluorescence images of mESC showing that SS18 (green) co-localizes with ELL3 (red) and EP300 (grey) in puncta. DNA was stained with DAPI (blue). Scale bar: 10 μm . **b.** Line scans of the images of a cell co-stained for SS18, ELL3 and EP300, at the position depicted by the red line. The direction is from the top left to the bottom right. **c,d.** FRAP experiments for SS18 (**c**) and ELL3 (**d**).

Left, representative images of the FRAP experiment. The white arrow refers to the punctum undergoing photobleaching. Right, quantification of FRAP data for puncta of SS18 ($n = 5$) and ELL3 ($n = 10$). The puncta were photobleached at $t = 0$ s and data are plot as means \pm standard error. **e-g**. GSEA-like analyses for SS18 (**e**), EP300 (**f**) and ELL3 (**g**), with all focus sites (CondSig-positive and -negative peaks of the CAP) were ranked by the \log_2 -transformed fold change in CUT&RUN signals and annotated against the set of CondSig-positive peaks. Left, 1,6-HD treatment versus wild type, right, 1,6-HD treatment versus 2,5-HD treatment, and the pseudo count in fold change calculations was set to 0.1. The GSEA-like analyses were performed with Python package GSEAPy. The enrichment score profile, the position of CondSig-positive sites and fold change profiles were shown, and the normalized enrichment score (NES) and false discovery rate (FDR) were labeled. **h**. UCSC genome browser view of representative CondSig-positive sites. Signals represent RPM and the related loci were shaded in purple.

4. Related to the previous point, 1,6-HD treatment is very harsh and controversial. It can have many undesirable effects. Introducing mutations that abrogate phase separation for well characterized proteins like MED1 (line 332) and testing effects on transcriptional bursting would be much more compelling.

Thank you for your insightful comments. According to your suggestion, we ablated the IDR of MED1 based on the annotation in MobiDB using CRISPR-Cas9, and then generated single-cell RNA-seq data in mutated mESC using 10X Genomics platform. However, upon inferring burst frequencies of genes from single-cell RNA-seq data, we observed no significant reduction in either the CondSig target genes or other genes after the ablation of MED1-IDR (Fig. R4). A potential explanation for this observation may be due to the off-target effects or editing efficiency variability among individual cells.

Fig. R4. Burst frequencies of CondSig target genes and other genes before and after the ablation of MED1-IDR in mESC. Stastical significance was evaluated by a two-sided Welch's t -test, and n.s. means non-significant.

We further explored this issue by using 2,5-HD treatment as a control for 1,6-HD treatment. Although 2,5-HD is chemically similar to 1,6-HD, it is not as strong as 1,6-HD in dissolving the hydrophobicity-dependent condensate (Lin *et al.*, Cell, 2016). We generated single-cell RNA-seq data in 2,5-HD-treated mESC and inferred burst frequencies. Using 2,5-HD treatment as the control for comparison, we still observed that target genes of CondSig displayed a significant reduction in burst frequencies upon 1,6-HD treatment (updated Fig. S10c).

We acknowledge the concern regarding the potentially harsh and undesirable effects of 1,6-HD treatment. To deeply investigate the effect of biomolecular condensation on burst frequencies, more stringent experimental approaches are required in future studies. We discussed this issue in the revised Discussion section.

5. The authors do not confirm the liquid-like properties of the puncta they find (FRAP). The puncta could be all kinds of assemblies.

According to your insightful suggestions, we conducted FRAP experiments to investigate the liquid-like properties of nuclear puncta, including SUPT6H, CTR9 and SUPT5H (Fig. R5, *i.e.*, updated Fig. 4c-e). In addition, as mentioned above, we performed FRAP experiments for our newly validated components CAPs: SS18, and ELL3 (Fig. R3c,d, *i.e.*, updated Fig. S9c,d). All these examined component CAPs showed rapid recovery after photobleaching.

Fig. R5 (*i.e.*, Fig. 4c-e). FRAP experiments for SUPT6H, CTR9 and SUPT5H in mESC. Left, representative images of the FRAP experiment. The white arrow refers to the punctum undergoing bleaching. Right, quantification of FRAP data for puncta of SUPT6H ($n = 6$), CTR9 ($n = 5$) and SUPT5H ($n = 5$). Puncta were photobleached at $t = 0$ s, and data were plot as mean \pm standard error.

6. Fig 5b and c, the pronounced effect on busting may be related to the histone modifications that are associated with the CondSig-positive sites. A proper control would be targets of CondSig-negative sites with the same histone modifications.

Thank you for the valuable feedback. According to your suggestion, we added a comparison between target genes of all CondSig-positive sites and all CondSig-negative sites (*i.e.*, sites occupied by at least two CAPs but not identified as CondSig-positive) with the same active histone modifications or chromatin accessibility. We observed that CondSig-positive target genes always showed higher burst frequencies compared to CondSig-negative targets (Fig. R6, *i.e.*, updated Fig. S10e, f). This analysis provided direct evidence to support that the effect of CondSig on transcriptional burst frequency was unrelated to the active histone modifications or chromatin accessibility.

Fig. R6 (*i.e.*, Fig. S10e, f). Burst frequencies of targets of CondSig-positive / negative sites with same histone modifications or chromatin accessibility. Statistical significance was evaluated by a two-sided Welch's *t*-test.

7. Fig 2d, only IDRs of the proteome should be used as control, not the entire proteome.

We appreciate your insightful suggestion. In the revised manuscript, we categorized identified component CAPs into three groups: all component CAPs, component CAPs that are not known PSPs, and component CAPs that are not IDPs. And we used all IDPs and the entire proteome as two sets of control. Our analysis demonstrated that the non-PSPs and non-IDPs groups of component CAPs showed comparable fractions of charged blocks to the IDPs group, which were significantly higher than the entire proteome (Fig. R7, *i.e.*, updated Fig. 2d).

Fig. R7 (*i.e.*, Fig. 2d). Charged amino acid block fractions of component CAPs, all IDPs and the entire proteome, statistical significances for comparison between component CAPs / all IDPs and the entire proteome were shown.

8. The authors classify CAPs as scaffolds or clients. I am not sure their classification based on PS-self and PS-part is correct. PS-self is for phase separation driven by homotypic interactions and PS-part for heterotypic interactions. However, the involved proteins may be considered as scaffolds in both cases.

Thank you for your valuable feedback. We agreed that PS-Self proteins can spontaneously self-assemble to form droplets, while PS-Part proteins interact with partners to undergo phase separation, so they should both be considered as scaffolds. We modified our descriptions in the revised manuscript and figures accordingly.

9. Stats: Lots of trends are assessed statistically. Are p values corrected for multiple testing?

During the revision, we adjusted our p -values for multiple testing using the Benjamini-Hochberg (BH) procedure, which controls the false discovery rate in multiple testing. We updated the adjusted p -values in the revised Fig. 3a and Fig. S5a.

Reviewer #3 (Remarks to the Author):

The manuscript from Yu and coworkers describes a new computational framework to predict signatures of proteins bound to the chromatin in a condensate-like manner. They use the resulting CondSigDetector framework to predict known and previously unknown putative chromatin-associated condensates. While the approach could potentially be interesting, several methodological and technical concerns have to be addressed to show the validity of the CondSigDetector as outlined below.

1. The authors state that DNA binding co-occupancy could not be explained by DNA motifs or chromatin accessibility. However, the presence of histone modifications or rather interactions with others proteins may provide some information here. The authors should try to assess this, before claiming that the most-likely alternative explanation lies in condensate formation (line 95-98).

Thank you for your insightful comment. According to your comment, we assessed the correlation between histone modifications and co-occupancy events of CAPs. We found that high histone modification signals could not predict co-occupancy events very well (Fig R8a,b, *i.e.*, updated Fig. S1e,f). We also examined the relationship between protein-

protein interactions and co-occupancy events of CAPs, and found that the protein-protein interaction frequency of co-occupancy events is moderate (Fig. R8c,d, *i.e.*, updated Fig. S1g,h). We fully agree that different potential mechanisms to explain the co-occupancy events of CAPs are not mutually exclusive, with condensate formation is one of them. To avoid any misunderstanding, we updated the description in the revised manuscript.

Fig. R8 (*i.e.*, Fig. S1e-h). **a,b.** ROC curve analysis for the correlation between epigenetic modification signals and the presence of co-occupancy events in mESC (**a**) and K562 (**b**). **c,d.** The fraction of co-occupancy CAPs with or without protein-protein interaction (PPI).

2. In general, how CondSigDetector works and what the underlying assumptions are, is not well explained and should be improved in a revised manuscript.

Thank you for your valuable feedback. In the Discussion section of the revised manuscript, we have enhanced the explanation for the CondSigDetector's principles and its underlying assumptions. CondSigDetector was designed based on the increasing evidences suggesting chromatin-associated biomolecular condensates are mediated by collaborative interactions of components at specific genomic loci. The key assumption it relies on is that specific collaborations among CAPs on the genome, particularly those involving proteins with high condensation potential (such as known PSPs and IDPs), may act as signatures of chromatin-associated biomolecular condensates. CondSigDetector

aims to identify specific co-occupancy patterns among CAPs and filter co-occupancy patterns with a high probability of involving biomolecular condensation, termed CondSigs. It further delineates genomic regions exhibiting such co-occupancy patterns as CondSig-positive sites.

3. The authors identify almost 50k sites that are supposedly involving CondSigs. This could easily correspond to half of all CREs in a given cell type. Is that more than expected? Are all sites then regulated by formation of condensate-like structures?

Thank you for your insightful comment. According to your comment, we examined the fraction of CondSig-positive sites among all CREs. By defining CREs as genomic loci occupied by at least two CAPs, we identified 234,476 CREs in K562 and 272,840 CREs in mESC. The fraction of CondSig-positive sites is 22.6% in K562 and 14.2% in mESC. Considering the increasing evidences for the widespread presence of chromatin-associated biomolecular condensates, these fractions might be reasonable.

Nevertheless, we agree that it is possible that not all CondSig-positive sites serve as condensate-related sites due to the dynamic property of condensates. We further filtered high-confidence condensate-related sites from CondSig-positive sites as follows. The high-confidence condensate-related sites are required to exhibit elevated concentrations of all component CAPs (i.e., requiring each component CAP to have more than a 1.5-fold increase in signals at condensate-related sites compared to its ChIP-seq peaks not associated with CondSigs). This approach yielded 16,010 and 15,303 high-confidence condensate-related sites in K562 and mESC, respectively. We provided more descriptions of this issue in the Discussion section of the revised manuscript.

4. The authors distribute regions into Condsig-positive and -negative sites, but how do these sites relate to each other as no information is given as to how the CondSig-negative regions are chosen? Further, if the positive regions have higher accessibility and thus a higher number of TFs bound, how fair is it to compare this to negative regions with lower accessibility? It would be better to have regions that are matched in terms of genomic distribution, size and accessibility and then to compare the performance of the CondSigDetector algorithm on this.

Thank you for your valuable suggestions. In performance evaluation section of our original manuscript, we divided the ChIP-seq peaks of each component CAP into CondSig-positive peaks or -negative peaks based on their overlap with genomic regions where the CAP was identified as a component of any CondSigs. This grouping was

intended to analyze whether the component exhibited a higher level of signal strength or chromatin contact frequency within CondSigs. According to your suggestion, we further refined the criteria for CondSig-positive and -negative peaks to ensure a fairer comparison as follows. We required that both refined CondSig-positive and -negative peaks were matched in terms of chromatin accessibility (with the presence of ATAC-seq peaks), or the number of co-occupied CAPs (with occupancy events of more than 10 CAPs). We also considered the effects of genomic distribution by analyzing promoter and non-promoter regions separately (*i.e.*, we compared CondSig-positive peaks in promoter regions to CondSig-negative peaks in promoter regions, and similarly for non-promoter regions). With this rigorous matching, we compared the levels of signal strength and chromatin contact frequencies between the refined CondSig-positive and -negative groups. The CondSig-positive groups consistently showed significantly higher signal strength and chromatin contact frequencies (updated Fig. S6, S7), supporting the robustness of CondSigDetector.

5. There is no evaluation on the CondSigDetector in term of accuracy on positive control sites for example (e.g. sites that are impacted in terms of accessibility / H3K27ac upon BRD4 inhibition).

Thank you for your valuable suggestions. Super-enhancers, recognized as large clusters of transcriptionally active enhancers, have previously been validated as biomolecular condensates at chromatin. Consequently, we collected super-enhancers from SEdb 2.0 (Wang *et al.* NAR, 2022), and found that 93.8% (743 out of 792) of super-enhancers in mESC and 97.5% (668 out of 685) in K562 overlapped with CondSig-positive sites. Such a high overlap supported the sensitivity of our prediction.

In addition, based on your suggestion, we evaluated the reliability of CondSig-positive sites by considering sites significantly affected by the inhibition of well-characterized CAPs involved in biomolecular condensates as potential positive markers. We re-analyzed H3K27ac, Pol II ChIP-seq data in mESC following the inhibition of EP300, an important chromatin regulator associated with biomolecular condensates in mESC (Ma *et al.* Mol Cell, 2021). After treatment with A-485, the EP300 inhibitor, we observed a substantial decrease in H3K27ac and Pol II signals at EP300 peaks. Specifically, 51.2% (3,652 out of 7,126) of CondSig-positive EP300 peaks exhibited a significant decrease (\log_2 -transformed fold change < -1) in H3K27ac signals, while only 20.8% (1,231 out of 5,906) CondSig-negative EP300 peaks exhibited a similar significant decrease (Fig. R9, *i.e.*, updated Fig. S4f). For Pol II, the percentages were 42.7% and 21.9%. The results indicate that transcription regulation of CondSig-positive sites was greatly affected by

EP300, a condensate-involved CAP, supporting the accuracy of identified CondSig-positive sites.

Fig. R9 (*i.e.*, Fig. S4f). The fraction of CondSig-positive or -negative EP300 peaks with decreased H3K27ac or Pol II signals after A-485 (EP300 inhibitor) treatment. The significant decrease was defined as \log_2 -transformed change (A-485 treatment versus wild type) < -1 .

6. How does the CondSigDetector perform when specific TFs, or specific TF classes, are omitted from the training of the algorithm?

We appreciate your suggestion regarding the robustness of CondSigDetector. According to your suggestion, we performed CondSigDetector prediction by excluding members of the CxxC (4 CAPs), bZIP (2 CAPs), bHLH (5 CAPs) and C2H2 (12 CAPs) families from the mESC dataset, as well as the corresponding four TF families comprising 1, 16, 10, and 22 CAPs from the K562 dataset. The prediction results after excluding specific TF families, recovered most of the initially identified CondSigs that were predicted with the full dataset (Fig. R10, *i.e.*, updated Fig. S2g,h), confirming the robustness of CondSigDetector prediction.

Fig. R10 (*i.e.*, Fig. S2g,h). The fractions of original CondSigs recovered by predictions with excluding specific TF families from the dataset. The name of family and the number of CAPs excluded from each TF family were labelled in X-axis.

7. In line 189-190, the authors state that a certain percentage of CAPs identified as CondSigs are scaffolds or scaffold-clients, but this may indirectly be art of the CondSigDetector categorization as related features such as co-occurrence in MLOs were used in the training. Same holds true for the DISP-seq, as CAPs with a higher percentage of IDRs will be higher valued in the CondSigDetector algorithm, thus this is a self-conformation in a way?

Thank you for your thoughtful comment. According to your comments, we have removed all known MLO memberships from the component CAPs to perform an additional evaluation. As another reviewer pointed that both PS-Self and PS-Part proteins should be defined as scaffolds, we used the terms PS-Self and PS-Part in the revised manuscript. As shown in the updated Fig. S4e, most remaining component CAPs exhibited both high PS-Self and PS-Part ranking scores, or in one of the two. This suggests the high phase separation capacities of our predictive components.

For the DisP-seq issue, a similar comment was also raised by another reviewer. We agreed that the binding sites of IDR-containing proteins (IDPs) are proposed to have higher DisP-seq signals. We further compared the DisP-seq signals at all CondSig-positive sites with a control group of sites occupied by any IDPs (termed as IDP-positive sites), and we found that all CondSig-positive sites exhibited higher DisP-seq signals than IDP-positive sites (Fig. R1). Nevertheless, to eliminate the possibility of circularity, we removed this part from our revised manuscript.

To enhance the validation of identified CondSigs, we utilized PSPire (Hou *et al.* Nat Commun, 2024), a recently developed phase-separating proteins (PSPs) predictor that incorporates both residue-level and structure-level features. After removing known PSPs from the component CAPs of identified CondSigs, the remaining component CAPs still exhibited higher PSPire scores compared to all IDPs predicted by MobiDB-lite or the entire proteome, indicating the high phase separation capacities of identified component CAPs (Fig. R2). We added the results as updated Fig. 2e.

8. In general, the authors make very strong claims while they only provide very limited data to support these. And thus some tuning down statements would highly benefit the credibility of the presented manuscript.

According to your suggestions, we performed more experiments and analyses to provide more evidences for our conclusion. Nevertheless, we agree that some claims require more experimental evidence. In the revised manuscript, we tuned down some claims in the Results section and added more discussions in the Discussion section.

9. The authors do only one experimental assay, using 1,6-hexanediol (1,6-HD). But this assay is very harsh, and proper controls should be included, for example 2,5-HD. And a lot more data should be shown besides a fold change and one genomic region. That makes any reader doubt how big the changes actually are. Where is an overview of total numbers of DE peaks, genome-wide overview, more example loci?

We appreciate your constructive feedback. According to your suggestion, we incorporated 2,5-HD treatment as a proper control in our CUT&RUN assays for DDX21, SUPT6H, and CTR9. Although 2,5-HD is chemically similar to 1,6-HD, it is not as strong as 1,6-HD in dissolving the hydrophobicity-dependent condensates (Lin *et al.*, Cell, 2016). We then conducted two types of comparative analyses: 1,6-HD treatment versus wild-type, and 1,6-HD treatment versus 2,5-HD treatment. To provide a genome-wide overview of changes in component CAP's signals under different comparisons, we performed a GSEA-like analysis for each CAP to quantify the enrichment level of their CondSig-positive peaks in genomic loci with decreased signals. We observed significant enrichments of CondSig-positive peaks in genomic loci with decreased signals both in two types of comparisons (Fig. R11a-c, *i.e.*, updated Fig. 4f,g, Fig. S8a,c,h,i). This evidence indicated that CondSig-positive peaks are more sensitive to 1,6-HD treatment and suggested their condensation properties. We also provided UCSC genome browser snapshots to show more example loci (Fig. R11d-f, *i.e.*, updated Fig. 4h, S8b,j).

Fig. R11 (*i.e.*, Fig. 4f,g,h, Fig. S8 a,b,c,h,i,j). **(a-c)** GSEA-like analyses for DDX21 **(a)**, SUPT6H **(b)** and CTR9 **(c)**, with all focus sites (CondSig-positive and -negative peaks of the CAP) were ranked by the \log_2 -transformed fold change in CUT&RUN signals and annotated against the set of CondSig-positive peaks. Left, 1,6-HD treatment versus wild type, right, 1,6-HD treatment versus 2,5-HD treatment, and pseudo count in fold change calculations was set to 0.1. The GSEA-like analyses were performed with Python package GSEAPy. The enrichment score profile, the position of CondSig-positive sites and fold change profiles were shown, and the normalized enrichment score (NES) and false discovery rate

(FDR) were labeled. (d-f) UCSC genome browser view of representative CondSig-positive sites. Signals represent RPM and the related loci were shaded in purple.

In addition, during the revision, we performed experimental validations for SS18, ELL3, and EP300. We performed CUT&RUN assays for these three CAPs after 1,6-HD treatment. Similarly, for these experiments, both wild type and 2,5-HD treatment were used as controls respectively (Fig. R3e-h, *i.e.*, updated Fig. S9e-h).

10. The authors frequently compare fold changes at CondSig-positive and negative regions, and claim higher fold changes as the positive regions. But these had for example higher signal to start with, so you would also get higher fold changes. So again, how fair is this comparison?

Thank you for your valuable feedback. In CUT&RUN data analysis, we assessed the fold changes of component CAP's signals between 1,6-HD treatment and controls at CondSig-positive and -negative peaks. We calculated fold change as follows:

$$FC = \log_2 \left(\frac{S_{treatment} + pseudo\ count}{s_{wt} + pseudo\ count} \right)$$

In this formula, wild-type signals serve as the denominator, so higher signals in control indeed correspond to lower fold changes with the same depletion level.

We understand your concern about the robustness of fold change calculations, and we implemented several strategies to improve the robustness. First, we normalized the reads counts by downsampling reads from different conditions to the equal total reads. Next, as shown in the formula, we included a pseudo count (set to 0.1 in this manuscript) in the fold change calculation to prevent extreme values from distorting our results. Moreover, it should be noticed that, the sites analyzed in this section, both CondSig-positive and -negative peaks, are ChIP-seq peaks of the given component CAP, which inherently exhibit signals significantly above the genomic background level, largely avoiding the risk of biased interpretations of fold changes. We provided more detailed descriptions of our data processing and fold change calculations in the Methods and Figure legends in the revised manuscript.

11. The authors talk about higher bursting at CondSig positive sites, but how do these relate to the CondSig negative sites in terms of genomic location, distance to genes, etc?

Thank you for your comments. In our original manuscript, genes whose promoter overlaps with any CondSig-positive sites, or which have long-range chromatin contacts with those sites, were defined as target genes. All other genes were used as the control. During the revision, to improve the robustness of our conclusions, we compared target genes of all CondSig-positive sites and all CondSig-negative sites (*i.e.*, sites occupied by at least two CAPs but not identified as CondSig-positive) with the same histone modifications or chromatin accessibility. We observed that CondSig-positive target genes always showed higher bursting frequencies compared to CondSig-negative targets with these different constraints for selecting target genes (Fig. R6, *i.e.*, updated Fig. S10e, f). We added the details of the target gene assignment in the Methods section.

12. Figure 5, what is genomic background?

In Fig. 5a, the genomic background used in showing the histone modifications enrichment at CondSig-positive sites is the average signal of all 1-kb bins in the genome. We added a detailed description in the Figure legend.

Reviewers' Comments:

Reviewer #2:

Remarks to the Author:

The authors addressed all my concerns.

Reviewer #3:

Remarks to the Author:

The authors have addressed most of my comments in an adequate manner. As a final comment: it is good to see that 2,5-HD is now included as a control. I do however find that showing only a few example loci (which could be cherry picked) and GSEA does not do complete justice to this data. It would be good as a final addition that the authors include heatmaps of the ChIP-seq signal in all conditions for the peaks that change by 1,6-HD compared to control. This could be all changed peaks and / or only those predicted by CondSig. And in addition, statistics that show that CondSig-predicted peaks are more often disrupted by 1,6-HD compared to 2,5-HD (plus the number of impacted peaks in each condition). This together is important to give the readers a better feeling on the magnitude of the effects in a genome-wide fashion.

Reviewer #2 (Remarks to the Author):

The authors addressed all my concerns.

Reviewer #3 (Remarks to the Author):

The authors have addressed most of my comments in an adequate manner. As a final comment: it is good to see that 2,5-HD is now included as a control. I do however find that showing only a few example loci (which could be cherry picked) and GSEA does not do complete justice to this data. It would be good as a final addition that the authors include heatmaps of the ChIP-seq signal in all conditions for the peaks that change by 1,6-HD compared to control. This could be all changed peaks and / or only those predicted by CondSig. And in addition, statistics that show that CondSig-predicted peaks are more often disrupted by 1,6-HD compared to 2,5-HD (plus the number of impacted peaks in each condition). This together is important to give the readers a better feeling on the magnitude of the effects in a genome-wide fashion.

According to your valuable suggestion, we added heatmaps displaying signals for DDX21, SUPT6H, CTR9, SS18, EP300 and ELL3 across all their changed peaks (including both decreased and increased peaks) in three conditions: wild type, 2,5-HD treatment, and 1,6-HD treatment (Fig. R1a). We further calculated the number and fraction of CondSig-positive / negative peaks with decreased signals in 1,6-HD treatment compared to 2,5-HD treatment, and performed Fisher's exact tests for each factor. As shown in Fig. R1b, there is a significant enrichment of CondSig-positive peaks in the group of decreased peaks for each factor. We added the results as updated Fig. S8d,e, Fig. S9g-j, and Fig. S11a-f. Taken together, our results demonstrate that CondSig-positive peaks are more often disrupted by 1,6-HD treatment compared to 2,5-HD treatment.

Fig. R1. Genome-wide occupancy of CAPs in wide-type, 2,5-HD treatment, and 1,6-HD treatment.

(a) Heatmaps showing CUT&RUN signals for DDX21, SUPT6H, CTR9, SS18, EP300 and ELL3 across their peaks with changed signals (including decreased and increased peaks) when comparing 1,6-HD treatment to 2,5-HD treatment. Peaks with decreased signals (\log_2 -transformed fold change < -0.25) were marked in dark blue and peaks with increased signals (\log_2 -transformed fold change > 0.25) were marked in purple. Each group of peak was ranked based on the result of a k -means clustering without predetermined cluster centroids, which utilized signals from both 2,5-HD treatment and 1,6-HD treatment. (b) Bar plots showing the fractions of CondSig-positive / negative peaks with decreased signals out of all CondSig-positive / negative peaks, the exact number of peaks was labelled on the top of bars. A Fisher's exact test was performed to test the significance of enrichment of CondSig-positive peaks in the group with decreased signals and p -value was explicitly labelled.

Reviewer #3 (Remarks on code availability):

I didn't test the code but it seems well documented, usable and easy to install. The only thing that may be slightly worrying is that it is indicated that FilterSig takes 1-2 days to complete and needs a lot of CPU, but this may be inherent to the approach and the way it is coded.

Thank you for your thorough review. Regarding your concern about the computational demands of the FilterSig module, the high CPU usage (10 CPU cores) and extensive running time are indeed necessitated by the iterative evaluation and filtration for 1,325 signatures in mESC and 1,147 signatures in K562. To address your concern, we have optimized the code to significantly reduce the running time to 8 hours for mESC signatures and 5 hours for K562 signatures, while still utilizing 10 CPU cores. These improvements make the new version of the software more efficient and user-friendly.

Reviewers' Comments:

Reviewer #3:

Remarks to the Author:

I have no further comments